# Greenhouse Gas Emissions Performance of Electric and Fossil-Fueled Passenger Vehicles with Uncertainty Estimates Using a Probabilistic Life-Cycle Assessment

Robin Smit * and Daniel William Kennedy

Transport Energy/Emission Research (TER), Brisbane, QLD 4068, Australia; info@transport-e-research.com
* Correspondence: robin.smit@transport-e-research.com

**Abstract:** A technology assessment is conducted for battery electric and conventional fossil-fueled passenger vehicles for three Australian scenarios and seven Australian states and territories. This study uses a probabilistic life-cycle assessment (pLCA) to explicitly quantify uncertainty in the LCA inputs and results. Parametric input distributions are developed using statistical techniques. For the 2018 Australian electricity mix, which is still largely fossil fuels based, the weight of evidence suggests that electric vehicles will reduce GHG emission rates by 29% to 41%. For the 'fossil fuels only' marginal electricity scenario, electric vehicles are still expected to significantly reduce emission rates by between 10% and 32%. Large reductions between 74% and 80% are observed for the more renewables scenario. For the Australian jurisdictions, the average LCA GHG emission factors vary substantially for conventional vehicles (364–390 g $CO_2$-e/km), but particularly for electric vehicles (98–287 g $CO_2$-e/km), which reflects the differences in fuel mix for electricity generation in the different states and territories. Electrification of the Tasmanian on-road fleet has the largest predicted fleet average reduction in LCA greenhouse gas emissions of 243–300 g $CO_2$-e/km. A sensitivity analysis with alternative input distributions suggests that the outcomes from this study are robust.

**Keywords:** motor vehicle; greenhouse gas emissions; battery electric; life cycle; LCA; Monte Carlo; bootstrap; truncated distribution; BEV; ICEV



## 1. Introduction

To properly assess greenhouse gas (GHG) emissions performance of different vehicle technologies, a holistic and systematic method is required that evaluates all aspects of a vehicle's life and its associated impacts (cradle-to-grave). Life-cycle assessment or LCA is a method used to quantify the environmental impacts of a product's manufacture, operational use and end-of-life [1]. LCA can help clarify potential trade-offs between different environmental impacts and between different stages of the life cycle [2]. The comprehensive scope of LCA is useful in avoiding problem shifting from one life-cycle phase to another, from one region to another, or from one environmental problem to another [3].

There are different types of environmental impacts that can be assessed with LCA such as GHG emission impacts, toxicity, mineral resource depletion and land use [4,5]. Given the complexity and detailed consideration of the various aspects of a vehicle's life, LCA studies often incur restrictions in scope. For instance, LCA studies can focus on specific environmental impacts (e.g., greenhouse gas emissions only) or have a broader consideration of environmental impacts but focus on a few specific vehicle makes/models [5,6]. LCA studies can be set up in different ways, naturally with several underlying assumptions. LCA considers processes that are complex, location specific and vary in time, as well as over time (trends). It is therefore not surprising that LCA studies have caused diverging arguments about the environmental performance of the technology that is assessed [4,7]. Significant differences in LCA results have been reported for similar electricity generation

technologies, reflecting differences in local conditions as well as differences in LCA methods and assumptions [3]. Given the complex, localized and dynamic nature of life-cycle impacts, it is important that the uncertainty in LCA results is quantified and that LCA results are regularly updated and improved.

Over the past 25 years or so, LCA has been deployed extensively to compare the environmental impacts of vehicles [8]. Of particular interest is the global move to electric vehicles with the aim to significantly reduce greenhouse gas emissions from road transport. Various studies have compared GHG life-cycle emissions of fossil-fueled internal combustion engine vehicles (ICEVs) for either (plug-in) hybrid electric vehicles (PHEVs or HEVs) [1,9–11], battery electric vehicles (BEVs) [5,12–16] or a wider range of vehicle technologies [3,4,6,7,17–22]. Although the majority of LCA studies have used deterministic approaches and may quantify uncertainty (or sensitivity) using a scenario modelling approach [1,5,9,10,14–20,22], a limited number of studies have deployed a probabilistic approach to LCA to explicitly account for substantial variability and uncertainty in input information [6,11,12,21]. Twenty years ago, one study [12] demonstrated the use of Monte Carlo simulation within a life-cycle framework. The authors stated: "A great deal of effort is spent debating the 'most appropriate' value to select for a given input variable to include how long a vehicle or battery will last or what the air emission factor should be. All these variables have uncertainty and variability associated with them. Monte Carlo simulation is a tool well suited to understand the magnitude of the uncertainties and variability that are difficult to observe using deterministic methods." It is noted that some variation in application of uncertainty analysis exists. For instance, one study used an uncertainty analysis as a separate exercise after a deterministic LCA was conducted [23]. In another study, probability distributions were used for one particular aspect of the LCA only [13].

There are benefits of using a probabilistic LCA approach as compared to a deterministic LCA [6,12]. It provides the decision maker with a range of potential and representative outcomes along with the predicted chance of their occurrence. pLCA facilitates incorporation of scarce input information and input information of varying or unclear quality that can be contradictory. It enables incorporation of expert judgement and can estimate the uncertainty based on a broad range of viewpoints. The approach is relatively fast using the best available information, without the need for compilation of often expensive and time-consuming input data. Moreover, study details are presented and accessible and the pLCA method is transparent. These points were all identified as general issues after a review of 51 EV life-cycle studies [8]. Finally, it guides improvement efforts towards specific aspects of the LCA assessment that matter, therefore providing clear direction to improve the accuracy of the LCA outcomes cost-effectively.

Nevertheless, the results of probabilistic LCA rely on assumptions underlying probabilistic input distributions, which are in turn affected by the availability of relevant information. In addition, the robustness of the results also depends on the formulation of the LCA model.

This study will deploy a probabilistic LCA approach to compare GHG life-cycle emissions of fossil-fueled internal combustion engine vehicles (ICEVs) and battery electric vehicles (BEVs) in Australia. It will demonstrate how this alternative approach results in a relatively rapid development of an LCA, which can be readily updated with new information and can progressively be refined and/or expanded, if so desired. In comparison with a limited number of previous probabilistic LCA studies, this study considers a wider range of possible distributions and includes the use of additional statistical techniques such as bootstrap resampling.

The focus of this study is on passenger vehicles in the Australian on-road fleet, which is quite unique and different from the commonly assessed US, EU or Asian fleets in terms of distributions of vehicle size and fuel type, emission standards and fuel quality [24–27]. For instance, a comparison with on-road fleets in the EU, USA and Japan confirms that new Australian passenger vehicles are distinctly different and underperforming in relation to $CO_2$ emission rates and fuel economy [25]. The results of this study will inform policy

makers about the current and future emission reduction potential of electrification of the transport sector. It is noted that there are many types of environmental impacts that can be assessed with LCA such as toxicity, mineral resource depletion, total life-cycle cost and land use [4,5,28], whereas the scope of this study is restricted to an assessment of GHG emission impacts.

## 2. Materials and Methods

### 2.1. Probabilistic LCA (pLCA)

A probabilistic LCA (pLCA) approach estimates uncertainty and variability in model predictions and quantifies non-linear interactions. Probabilistic analysis yields quantitative insight into both the possible range and the relative likelihood of values for model outputs [29]. The method is particularly useful to determine the robustness of study outcomes, which is important when comparing different technology options. Probabilistic LCA can also be used to identify which aspects of the LCA are most uncertain and warrant further targeted examination. This assists with cost-effective use of available resources to further improve LCA results.

The results of a probabilistic analysis rely critically on the probabilistic definitions of model input variables. These definitions are affected by limitations on the availability and the quality of available information and data. In this study, the probabilistic definition of input variables is based on statistical analysis of empirical data and results from peer-reviewed scientific studies, wherever available.

### 2.2. Model Definition

The life-cycle GHG emission factor is used as the assessment variable (functional unit). This variable normalizes the amount of GHG emissions per vehicle kilometer driven and is expressed as $CO_2$-e/vehicle km. Carbon dioxide equivalent ($CO_2$-e) emissions are computed by multiplying emissions of a particular greenhouse gas with its Global Warming Potential (GWP) and taking the sum of these emissions. Five GHG emission life-cycle aspects are considered: (1) production of the vehicle (manufacturing of non-battery components, manufacturing of the BEV battery), (2) production of (fossil) fuels for ICEVs (extraction, transport and fuel refining), (3) production of electricity for BEVs (extraction and transport of fossil fuels, electricity generation, electricity distribution losses and power generation infrastructure), (4) on-road operation or use of the vehicle (ICEV fossil-fuel use, BEV energy use, and BEV battery charging losses) and (5) disposal and recycling of the vehicle at the end of its life.

The life-cycle GHG emission factors $e_{ICEV}$ and $e_{BEV}$ are computed with two additive models and sub-models (if applicable). In Equations (1) and (3), $e_{i,j}$ is used to represent a GHG emission factor ($CO_2$-e/km) for life-cycle aspect I and vehicle type j.

$$e_{ICEV} = e_{vehicle,ICEV} + e_{infra,ICEV} + e_{upstream,ICEV} + e_{road,ICEV} + e_{disposal,ICEV} \tag{1}$$

$$e_{vehicle,ICEV} = W_{ICEV} \, \varphi_{v,ICEV} / M \tag{2}$$

$$e_{BEV} = e_{vehicle,BEV} + e_{infra,BEV} + e_{upstream,BEV} + e_{road,BEV} + e_{disposal,BEV} \tag{3}$$

$$e_{vehicle,BEV} = ((W_{BEV} - W_{BAT}) \, \varphi_{v,BEV} + \theta_b \, \varphi_b) / M \tag{4}$$

$$e_{infra,BEV} = \varepsilon \, \sigma_s / (\eta_g \, \eta_b) \tag{5}$$

$$e_{upstream,BEV} = \varepsilon \, \phi_s / \eta_b \tag{6}$$

$$e_{road,BEV} = \varepsilon \, \omega_s / (\eta_g \, \eta_b) \tag{7}$$

where $W_{ICEV}$ = ICEV vehicle weight (kg), $W_{BEV}$ = BEV vehicle weight (kg), $W_{BAT}$ = BEV battery weight (kg), $\varphi_{v,ICEV}$ and $\varphi_{v,BEV}$ are the respective carbon intensities of vehicle production (g $CO_2$-e/kg vehicle), $\varphi_b$ = carbon intensity battery production (g $CO_2$-e/kWh battery capacity), $\theta_b$ = battery capacity (kWh), M = lifetime vehicle mileage, $\sigma_s$ = GHG emission intensity electricity infrastructure (g $CO_2$-e/kWh generated) for scenario s, $\phi_s$ = GHG

emission intensity upstream fuels for electricity generation (g $CO_2$-e/kWh consumed 'at the power point') for scenario s, $\omega_s$ = GHG emission intensity electricity generation (g $CO_2$-e/kWh generated) for scenario s, $\eta_g$ = grid transmission efficiency (−), $\eta_b$ = battery recharging efficiency (−) and $\varepsilon$ = real-world electricity use BEV (kWh/km).

This study investigates fleet average impacts. Therefore, fleet averaged input data are used, such as mean vehicle weight and the associated probability distribution of this mean value. The pLCA method can similarly be applied to specific vehicles, if so desired, but this is beyond the scope of this study. This would require the use of vehicle-specific input data (for instance, Tesla Model 3 battery capacity and weight), rather than fleet averaged input information.

*2.3. Input Distribution Development*

The model variables in Equations (1) and (3) are defined as parametric distributions, which represent the probabilities of all possible values in sample space [29]. A probability model is typically represented mathematically as a probability distribution in the form of either a probability density function (PDF) or cumulative distribution function (CDF) with associated parameters (scale, shape, minimum, maximum, etc.).

Quantitative data were used when available to develop the input distributions, supplemented with information from the available scientific literature. Quantitative data are either empirical data, reported data in the scientific literature or software output. A number of statistical techniques were used to develop the input distributions, namely bootstrap analysis, Monte Carlo simulation and parametric distribution fitting.

For LCA aspects where empirical input data were available, the data were used either directly as sampling distributions or sampling distributions were developed using bootstrap analysis. The statistical bootstrap technique produces resampled input distributions for a statistic of interest—for instance, the mean or median [30]. The bootstrap simulation creates an approximate but asymptotically accurate sampling distribution from the original data through repeated resampling with replacement and calculation of the statistic of interest [31]. From this distribution (non-symmetric), standard errors and confidence intervals are typically derived. The boot R package was used to perform the bootstrap analysis [32].

The sampling distributions were used to fit truncated parametric distributions by maximum likelihood [33]. They include the following candidates: Uniform (**U**: a, b), Triangular (**T**: a, b, c), Normal (**N**: m, s), Lognormal (**L**: m, s), Weibull (**W**: s, s), Gamma (**G**: s, r) and Exponential (**E**: s). The non-standard beta distribution (**B**: s, s) and the skew t-distribution (**S**: m, s, a, df) were also included to allow for additional flexibility in the fitting process [34,35]. The Dirac Delta function (**D**: m) is used to describe a constant value. Appendix A provides further information regarding the range, parameters and PDFs of the distributions. The location-scale t-distribution was also offered as candidate, but was not selected as the best fit throughout the analysis. Truncation is applied to the fitted distributions using the 'truncdist' R package by setting a lower limit a and an upper limit b [36]. This explicit definition of a plausible range prevents the use of unrealistic values in the pLCA. The R packages 'fitdistr' and 'fitdistrplus', 'extraDistr', 'sn' and 'truncdist' were used in the optimized fitting process [34,36–38].

The most appropriate parametric distribution was determined by comparing all fitted parametric distributions with the sampling data input values. This was performed visually using quantile–quantile (QQ) plots for all fitted distributions and by applying the Cramer Von Mises test statistic [39]. A QQ plot is a graphical method for comparing two probability distributions by plotting their quantiles against each other.

For model aspects where insufficient quantitative input data were available, two simplified distributions were used and parameters were estimated based on literature review. The uniform (rectangular) distribution is a probability distribution, which is simply defined by a lower limit a and an upper limit b of a plausible range. A uniform distribution represents equal probability between two end points. This distribution is appropriate if

only information on the lower and upper limit values are available [40]. The triangular probability distribution (T: a, b, c) is a continuous probability distribution, which is defined by a lower limit a and an upper limit b of a plausible range, as well as the most plausible estimate c. The triangular distribution is appropriate for situations in which the exact form of a distribution is not precisely known, but in which values toward the middle of the range of possible values are considered more likely to occur than values near either extreme [41]. The triangular probability distribution can be asymmetrical.

In a Monte Carlo simulation, random samples are taken from input distributions many times (100,000–1,000,000) and propagated through the appropriate model to create probability output distributions [42]. This way, not only are expected values estimated, but also the associated variability and uncertainty. The process is a mathematical analogue of an experiment, which is repeated many times to provide an accurate description of the variability in the output estimate. Monte Carlo simulation is applied in two different ways. First, it is used to combine different sampling distributions and create an output sampling distribution for the GHG emission factor ($CO_2$-e/km) for a particular life-cycle aspect and vehicle type. Second, Monte Carlo simulation is used to propagate the uncertainty and variability reflected in the parametric input distributions to the model outputs $e_{ICEV}$ and $e_{BEV}$. The output PDFs express both central tendencies and the variability in the output variables arising from the variation in the input variables. Uncertainty in the outputs is defined as a 95% confidence interval (CI) of the mean value and is stated as a value range (asymmetric confidence interval) or a percentage (symmetric confidence interval).

### 2.4. Scenario Definitions

Some model variables are inter-dependent (for instance, infrastructure and upstream fuel). These dependencies are accounted for by defining three scenarios for Australia with different input distributions. In addition, Scenario 1 (current situation) is further detailed by Australian state and territory (Figure 1). Using data from the Australian Energy Statistics [43], Table 1 presents the percentage of electricity generated by fuel type.

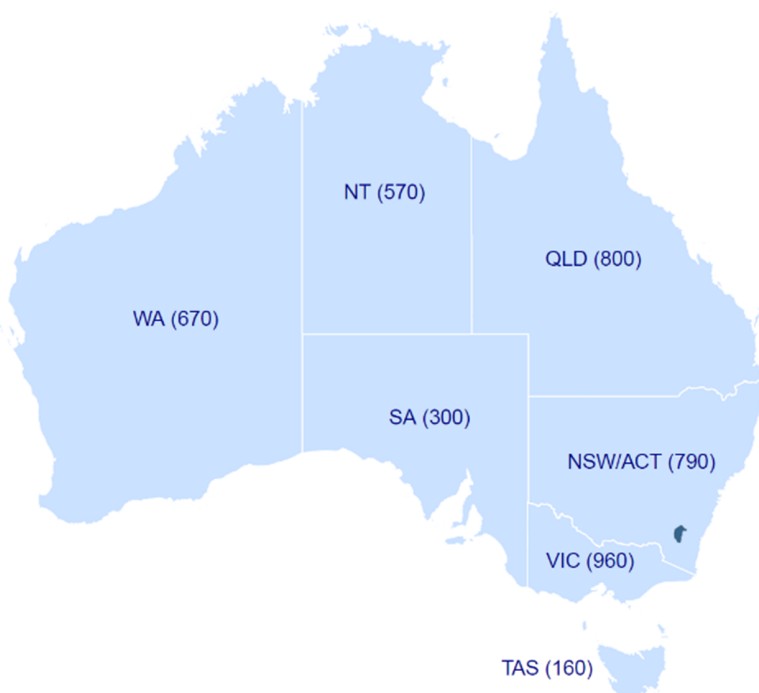

**Figure 1.** Scope 2 GHG emission intensities (g $CO_2$-e/kWh consumed) for electricity consumption in Australia by jurisdiction (WA = Western Australia, NT = Northern Territory, SA = South Australia, QLD = Queensland, NSW = New South Wales, ACT = Australian Capital Territory, VIC = Victoria, and TAS = Tasmania).

**Table 1.** Percentage of electricity generated by fuel type for each scenario or jurisdiction [24].

| Scenario, Jurisdictions | Coal | Gas | Oil | Nuclear | Hydro | Wind | Biomass | Solar |
|---|---|---|---|---|---|---|---|---|
| Australia Current (SC1) | 58.4% | 20.0% | 1.9% | 0.0% | 6.0% | 6.7% | 1.3% | 5.6% |
| Australia Marginal Electricity (SC2) | 73.0% | 24.0% | 3.0% | 0.0% | 0.0% | 0.0% | 0.0% | 0.0% |
| Australia More Renewable (SC3) | 5.0% | 5.0% | 0.0% | 0.0% | 30.0% | 25.0% | 5.0% | 30.0% |
| New South Wales (NSW) Current | 80.7% | 3.3% | 0.5% | 0.0% | 3.0% | 5.2% | 1.6% | 5.7% |
| Victoria (VIC) Current | 70.8% | 6.8% | 0.4% | 0.0% | 5.6% | 10.0% | 1.5% | 4.9% |
| Queensland (QLD) Current | 73.8% | 14.1% | 1.4% | 0.0% | 1.5% | 0.6% | 1.9% | 6.8% |
| Western Australia (WA) Current | 23.8% | 61.5% | 5.8% | 0.0% | 0.5% | 4.4% | 0.3% | 3.7% |
| South Australia (SA) Current | 0.0% | 48.5% | 1.1% | 0.0% | 0.1% | 38.2% | 0.6% | 11.5% |
| Tasmania (TAS) Current | 0.0% | 5.3% | 0.2% | 0.0% | 83.2% | 9.7% | 0.2% | 1.4% |
| Northern Territory (NT) Current | 0.0% | 78.6% | 17.8% | 0.0% | 0.0% | 0.0% | 0.2% | 3.4% |

Scenario 1 (SC1) reflects the Australian Electricity mix and on-road passenger vehicle fleet in the 2018–2019 financial year (1 July 2018–30 June 2019). Australia uses more fossil fuels than many other countries such as the EU, USA, Canada, Japan, India, China, South Korea, Russia and Brazil [44]. At the other end of the spectrum, Norway currently uses mainly renewable energy (98%), and is an example of what Australia could be like after transformation to a sustainable energy system is completed.

Scenario 2 (SC2) is a 'marginal electricity' scenario, which is 100% fossil fueled and assumes an Australian electricity mix of 73% coal, 24% gas and 3% oil. The use of average GHG emissions from electricity generation (SC1) may produce misleading results and it has been suggested that the use of marginal emissions from electricity generation is more accurate [15,45]. The marginal grid fuel mix typically has a higher emissions intensity than the average grid mix. Marginal electricity production reflects emissions from fossil-fueled power plants, which may be turned on to meet new demand from EV charging. Renewable energy sources are generally fully utilized and will not change their generation output in the short term when BEV penetration increases. In the short term, primarily coal and natural gas plants may increase generation in response to new loads, which is reflected in this scenario.

Scenario 3 (SC3) is an Australian 'More Renewable Energy' scenario, which assumes an Australian electricity mix of 5% coal, 5% gas, 30% hydro, 25% wind, 5% biomass and 30% solar. A more renewable Australian electricity grid mix has a substantially lower emissions intensity than the current largely fossil fuel-based grid mix, as will be discussed later.

## 3. Input Distributions

This section describes in detail how the parametric input distributions for the technology assessment are developed.

### 3.1. Overview of Input Distribution Definitions

Table 2 presents an overview of the parametric input distributions used in the probabilistic technology assessment. The bootstrap sampling distributions and fitted parametric distributions are shown in Figure 2. A detailed discussion regarding the development of the parametric input distributions in Table 2 is provided in the following sections.

### 3.2. Vehicle Manufacturing

GHG emissions per vehicle produced depends on make/model and manufacturing location, and more generally on type of materials used, vehicle size and weight and emission intensity of the energy used in vehicle production. For electric vehicles, an important aspect is battery production, which produces significant amounts of GHG emissions. The parametric input distributions for life-cycle GHG emission factors for vehicle manufacturing ($e_{vehicle,ICEV}$ and $e_{vehicle,BEV}$) were developed by defining the input distributions for models 2 and 4 (Section 2.2) as follows.

**Table 2.** GHG emission factor (g $CO_2$-e/km) input distribution definitions.

| Life-Cycle Aspect * | Vehicle Technology | LCA Model Input Variable | Distribution | Typical Value | Plausible Min–Max Value |
|---|---|---|---|---|---|
| P | ICEV | $e_{vehicle,ICEV}$ | Triangular, **T** (40.38, 44.98, 58.61) | 45.00 | 40.00–59.00 |
| P | BEV | $e_{vehicle,BEV}$ | Non-standard beta, **B** (7.30, 8.73) | 59.00 | 39.00–83.00 |
| I | ICEV | $e_{infra,ICEV}$ | Uniform, **U** (0.20, 2.50) | 1.30 | 0.20–2.50 |
| I | BEV | $e_{infra,BEV}$ | Non-standard beta, **B** (5.81, 10.44) ** (a) | 5.07 | 0.74–10.76 |
| U | ICEV | $e_{upstream,ICEV}$ | Uniform, **U** (35.90, 72.00) | 51.40 | 35.90–72.00 |
| U | BEV | $e_{upstream,BEV}$ | Lognormal, **L** (2.53, 0.53) ** (b) | 14.18 | 1.00–49.00 |
| O | ICEV | $e_{road,ICEV}$ | Normal, **N** (265, 3) ** (c) | 265.00 | 259.00–272.00 |
| O | BEV | $e_{road,BEV}$ | Non-standard beta, **B** (5.81, 10.44) ** (d) | 175.00 | 142.00–215.00 |
| D | ICEV | $e_{disposal,ICEV}$ | Uniform, **U** (0.10, 2.00) | 0.50 | 0.20–2.50 |
| D | BEV | $e_{disposal,BEV}$ | Uniform, **U** (0.10, 2.00) | 0.50 | 0.20–2.50 |

* P = production; I = infrastructure; U = upstream (Fuels); O = operation; D = disposal. ** Distribution definition varies and depends on scenario or jurisdiction; Scenario 1 is shown here as an example. (a) Refer to refer to below table: Infrastructure GHG emission factor (g $CO_2$-e/km) distribution definitions for Australian BEVs by scenario or jurisdiction. (b) Refer to below table: Upstream GHG emission factor (g $CO_2$-e/km) distribution definitions for Australian BEVs by sce-nario or jurisdiction. (c) Refer to below table: On-road GHG emission factor (g $CO_2$-e/km) distribution definitions for Australian ICEVs by scenario or jurisdiction. (d) Refer to below table: On-road GHG emission factor (g $CO_2$-e/km) distribution definitions for Australian BEVs by scenario or jurisdiction.

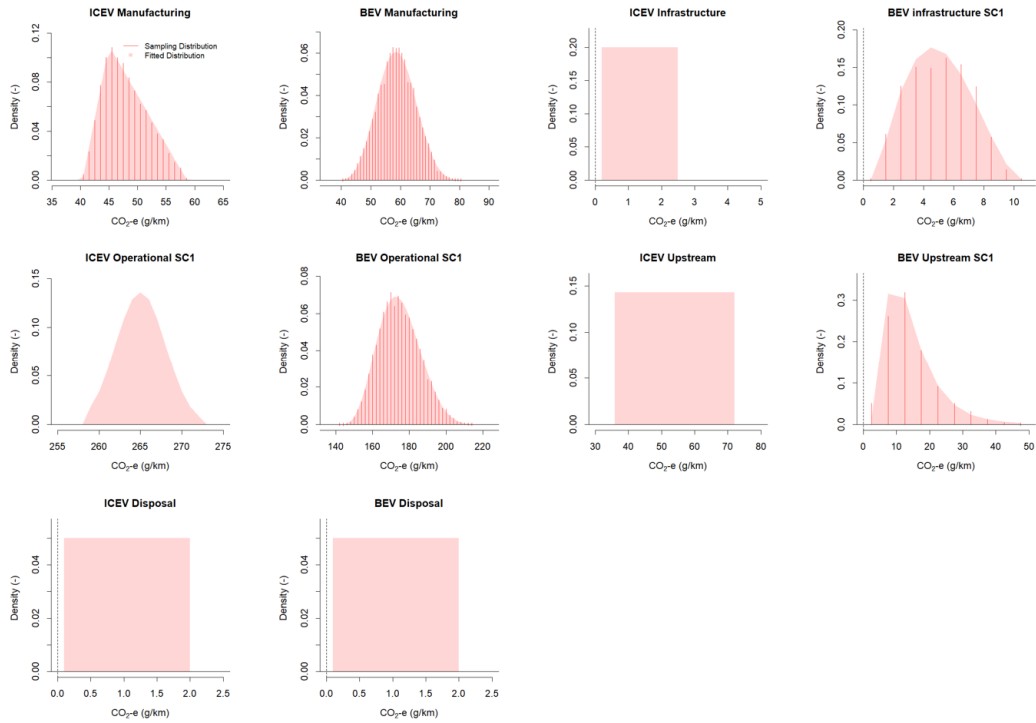

**Figure 2.** Parametric GHG emission factor (g $CO_2$-e/km) distributions by life-cycle aspect and technology group (bootstrap sampling distribution = vertical lines; fitted parametric distribution = shaded polygon).

For ICEV production, a plausible range for GHG emission intensity is 4.0–6.5 kg $CO_2$-e/kg of passenger vehicle, with a typical value of 5 kg $CO_2$-e/kg vehicle [46], (**T**: 4.0, 6.5, 5.0). The average weight of an Australian passenger vehicle is 1800 kg, with an estimated uncertainty (95% confidence interval) of 1% [24] (**U**: 1783, 1817). A study into worldwide BEV characteristics (n = 218) reported an average BEV vehicle mass of 1689 kg with an uncertainty of ±4% [47] (**U**: 1625, 1753). The fleet average weight for Australian BEVs is comparable with a value of 1600 kg [26]. GHG emissions for battery production need to be estimated separately and added. Battery manufacturing emissions are likely to fall between 41 and 156 kg $CO_2$-e per kWh of battery capacity, with a current average of approximately

100 kg $CO_2$-e per kWh [5,48] (**T**: 41, 156, 100). Average worldwide BEV battery capacity is estimated to be 46 kWh with an uncertainty of 8% [47] (**U**: 42, 50). BEV weight is corrected for the weight of the battery. A plausible range for battery energy density is assumed to be 0.12–0.16 kWh per kg of battery [7,49–51]. Using the plausible range in BEV battery capacity of 42–50 kWh, average battery weight is estimated to be 335 kg, which is 20% of total BEV vehicle weight (**T**: 265, 413, 335).

LCA studies have typically used lifetime PV mileage between 150,000 and 200,000 km, and even up to 320,000 km [21]. Although there were initial doubts about the durability of BEV batteries, batteries retain more than 90% of the original capacity beyond 200,000 km [5]. Normalizing for a lifetime mileage of 200,000 km and combining the input distributions through a Monte Carlo simulation (Equations (2) and (4)), production of an average Australian passenger vehicle is estimated to produce, on average, 48 g $CO_2$-e/km for ICEVs (plausible range 40–59) and 59 g $CO_2$-e/km for BEVs (plausible range 39–83). A triangular and non-standard beta distribution provides the best maximum likelihood fit to the sampling distributions for ICEV and BEV manufacturing, respectively (Table 2 and Figure 2). The results suggest that for the Australian market, BEV production produces on average approximately 20% more GHG emissions as compared with conventional fossil-fueled passenger vehicles, adding approximately 10 g $CO_2$-e per km to total life-cycle emissions for BEVs. Previous studies have used 35 to 46 g $CO_2$-e/km for ICEVs and 37 to 95 g $CO_2$-e/km for BEVs [4,15,46,48]. This study estimates a higher carbon footprint for Australian ICEVs, which is explained by the large proportion of large and heavy fossil-fueled passenger vehicles, as compared to, for instance, the EU market. The estimate for BEVs is within the ranges reported in other studies mentioned previously.

### 3.3. On-Road Driving ICEVs

The Australian Fleet Model (AFM) was used to create an input file for the vehicle emissions software COPERT Australia, reflecting the Australian on-road fleet for base year 2018. AFM is a fleet turnover simulation software that estimates the on-road vehicle population and total (vehicle) kilometers travelled (VKT) for 1240 vehicle classes for a prespecified base years [27]. The COPERT Australia software v1.3.5 was used to create an Australian motor vehicle emission inventory for base year 2018. The software predicts emissions for 226 Australian vehicle classes and accounts for the effects of driving behavior, meteorology and fuel quality [52–54]. COPERT Australia predicts total GHG emissions from road transport in 2018 as 85.1 million metric tons of $CO_2$-e, which is close to 85.2 million tons of $CO_2$-e reported by the Australian Greenhouse Emissions Information System (AGEIS) [55], a difference of 0.1%.

An average fuel consumption rate of 80.4 g/km (10.7 L per 100 km) is predicted for passenger vehicles by AFM/COPERT Australia. The corresponding GHG emissions factor for the on-road passenger vehicle (PV) fleet in 2018 is 257 g $CO_2$-e/km. The GHG to fuel ratio is 3.192. Analysis of the AFM/COPERT Australia results shows that average GHG emission rates are 247 g $CO_2$-e/km for petrol vehicles and 318 g $CO_2$-e/km for diesel vehicles. Diesel passenger vehicles have GHG emissions per kilometer that are 28% higher than their petrol counterparts. A recent study found that the main reason for this is that Australian diesel PVs are, on average approximately 40% heavier than petrol PVs [24]. Other diesel vehicle design parameters also adversely affect $CO_2$ emission rates, including a higher proportion of 4WD vehicles, 15% higher engine capacity and a low portion of CVT transmissions [24]. A (weighted) bootstrap analysis [31] using travel (VKT) by vehicle class as weights, estimates an uncertainty in this fleet average emission factor (ICEV) of approximately ±15%. These bootstrap results are similar to the reported uncertainty in fuel consumption of Australian PVs by the Australian Bureau of Statistics (ABS) of 5% to 11% [56]. A non-standard beta distribution (**B**: 9.89, 16.86) provides the best fit to the bootstrap sampling distribution of the fleet average emission factors with truncation at 225 and 298 g $CO_2$-e/km.

The ABS publishes the Survey of Motor Vehicle Use or SMVU [56], which includes a time-series of average rate of fuel consumption (liters/100 km) by jurisdiction and vehicle type. The ABS also reports the relative standard error (RSE). The standard error is a measure of the spread of estimates around the "true value" and RSE is the standard error that is expressed as a percentage of the estimate. The plausible range is defined as the 99.7% confidence interval (assuming a normal distribution), which is estimated as ±3 RSE. Fuel consumption (FC) and RSE data were retrieved from the SMVU [56] for passenger vehicles and for each jurisdiction. The average fuel density was calculated for each jurisdiction from total reported fuel consumption in mass and volume units in the SMVU. Average fuel density is used to convert units in L/100 km to g/km. Fuel consumption in g/km was then converted to g $CO_2$-e/km by multiplication of a factor of 3.192 mentioned earlier.

The average fuel consumption varies from 10.6 L/100 km (VIC) to 11.6 L/100 km (WA), which corresponds to 81 to 89 g fuel/km and 260 and 283 g $CO_2$-e/km, respectively. The highest fuel consumption rate of 90 g/km (11.3 L/100 km) and corresponding GHG emission factor of 286 g $CO_2$-e/km is reported for NT due to the high proportion of diesel use in this jurisdiction and resulting higher fuel density (Table 1). The plausible range for truncation varies between ±7% and ±14%, depending on the jurisdiction. The relative uncertainty in the converted ABS figures is assumed to be ± 3 RSE and follow a truncated normal distribution ($N$: 3.192 × FC, RSE × 3.192 × FC). For Australia as a whole, the uncertainty is reportedly smaller (±4%) and defined with a truncated normal distribution for the GHG emission factor ($N$: 265, 3). The distributions, typical values and truncation limits are shown in Table 3.

**Table 3.** On-road GHG emission factor (g $CO_2$-e/km) distribution definitions for Australian ICEVs by scenario or jurisdiction.

| Fuel Type | Distribution | Typical Value | Plausible Min–Max Value |
|---|---|---|---|
| Scenario 1 (Australia 2018/19) | Normal, $N$ (265, 3) | 265 | 256–275 |
| Scenario 2 (Marginal Electricity) | Normal, $N$ (265, 3) | 265 | 256–275 |
| Scenario 3 (More Renewable Electricity) | Normal, $N$ (265, 3) | 265 | 256–275 |
| NSW | Normal, $N$ (264, 7) | 264 | 242–286 |
| VIC | Normal, $N$ (260, 7) | 260 | 240–279 |
| QLD | Normal, $N$ (271, 7) | 271 | 249–293 |
| SA | Normal, $N$ (260, 7) | 260 | 241–280 |
| WA | Normal, $N$ (283, 7) | 283 | 262–303 |
| TAS | Normal, $N$ (265, 8) | 265 | 240–289 |
| NT | Normal, $N$ (286, 13) | 286 | 247–326 |

The GHG emissions factor for the Australian on-road passenger vehicle (PV) fleet derived from the SMVU (265 g $CO_2$-e/km) is 3% higher than the value predicted by COPERT Australia/AFM (257 g $CO_2$-e/km). The SMVU based GHG emission factor and associated uncertainty are used in the probabilistic technology assessment as this method enables prediction of these emission factors for all scenarios or jurisdictions. The results from the COPERT Australia/AFM method will be used to test the sensitivity of the study outcomes. The input distributions for on-road ICEV GHG emission factors are shown in Table 3 and Figure 3.

### 3.4. Electricity Production and Consumption

Indirect emissions due to electricity generation need to be estimated to quantify GHG emissions for BEVs. The Australian Energy Statistics (AES) provide data on the fuel mix used for electricity generation, which includes fossil-fueled power plants and generation by households and businesses [43,57]. In 2018/19, Australia's total electricity production of 263 TWh was mostly produced with fossil fuels (coal 60%, natural gas 20%) and 19% was produced with renewable energy sources (solar, wind, hydro, biomass), as was shown in Table 1.

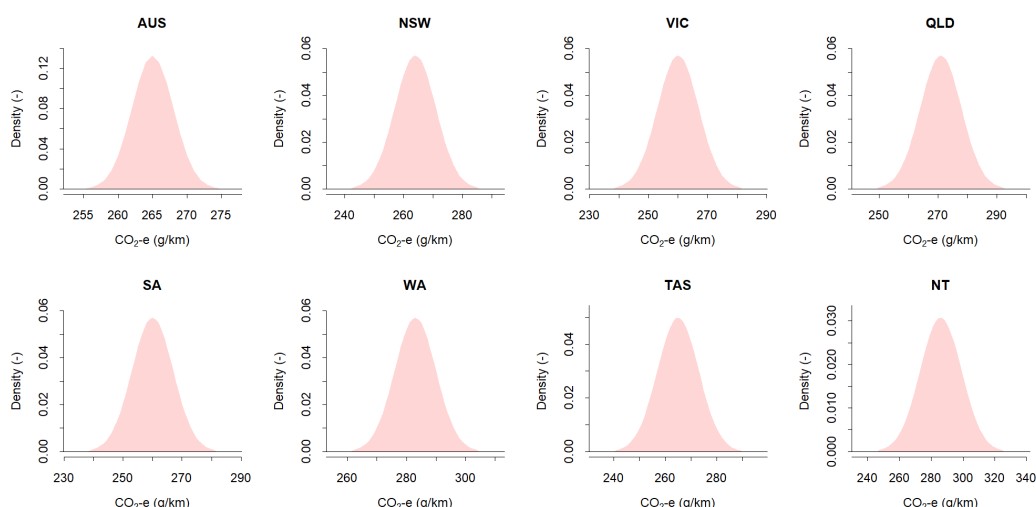

**Figure 3.** On-road GHG emission factor (g CO$_2$-e/km) distributions for Australian ICEVs by scenario or jurisdiction.

The National Greenhouse Accounts (NGA) provide 'Scope 2' GHG emission intensities for electricity consumption in Australia and for each state and territory [58]. The GHG emission intensity reflect the electricity fuel mix (fossil fuels, renewables) and include grid transmission losses. For Australia, the average GHG emission intensity for electricity production is reported to be 760 g CO$_2$-e/kWh for the 2018–2019 financial year. However, the values vary between jurisdictions with 160 g CO$_2$-e/kWh for Tasmania to 960 g CO$_2$-e/kWh for Victoria, as is shown in Figure 1.

The uncertainty in the NGA figures is not published. It is therefore assumed that the uncertainty is similar to the uncertainty in total fuel consumption by jurisdiction reported by the Australian Bureau of Statistics [56]. Similar to Section 3.3, the plausible range is defined as the 99.7% confidence interval, which corresponds to ±3 RSE reported by the ABS. The reported 99.7% CI varies between ±10 and ±13%, depending on the jurisdiction. The relative uncertainty in the published NGA figures is assumed to be ±15% and follow a truncated normal distribution, N (NGA, 0.05 NGA). For Australia the uncertainty is reportedly smaller (±4.7%) with a corresponding truncated normal distribution, N (NGA, 0.015 NGA). The input distributions for electricity consumption are shown in Figure 4.

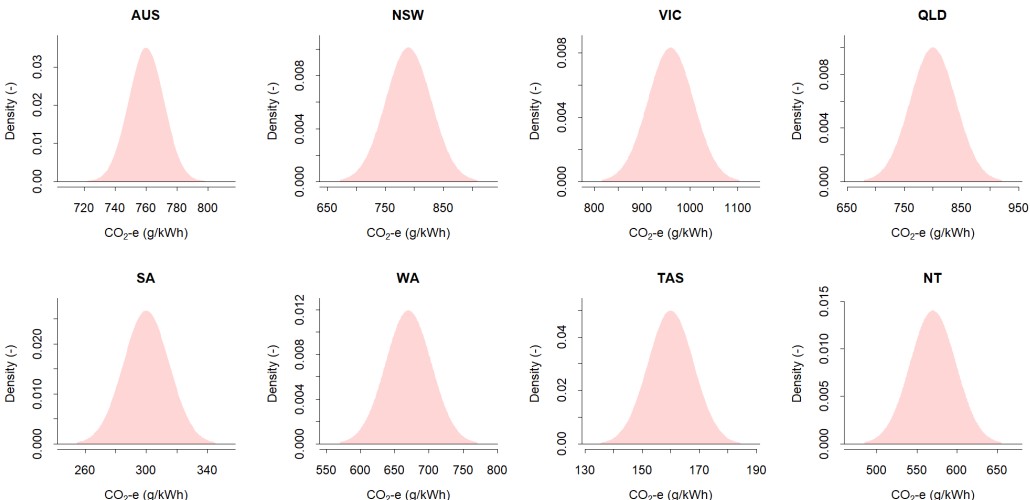

**Figure 4.** GHG emission intensity distributions for electricity consumption in Australia by jurisdiction (AUS = Australia, NSW = New South Wales, VIC = Victoria, QLD = Queensland, SA = South Australia, WA = Western Australia, TAS = Tasmania, and NT = Northern Territory).

To estimate GHG emission intensities for Scenario 2 (marginal electricity) and Scenario 3 (more renewables), industry data were collected and analyzed. Industry reports electricity production and Scope 1 and 2 GHG emissions to the Clean Energy Regulator or CER [59], which includes fossil fuels, as well as renewables. The data are published at generation facility level but exclude generators that are too small to report under the National Greenhouse and Energy Reporting Act 2007 (NGER Act). The CER data were collected for the 2018–2019 financial year (n = 345 grid-connected facilities) and used to estimate GHG emission factors and the associated variability and uncertainty for each energy source (biomass, coal, gas, hydro, oil, solar, wind). A (weighted) bootstrap analysis of the CER data shows that the average grid-connected emission intensity for Australia is 730 g $CO_2$-e/kWh generated, which is 4% lower than the NGA factor (760 g $CO_2$-e/kWh) likely due, in part, to grid losses that are not yet reflected.

Consumed electricity needs to account for energy losses due to transmission and conversion of electricity (grid losses). A plausible range for transmission and conversion losses is estimated to be 5–10%, with a typical value of 6% [45,60,61]. Note that efficiency is computed as 100% minus loss (%). The (weighted) bootstrap analysis was repeated for the seven fuel types. The bootstrap data were combined with a uniform input distribution for grid losses (**U**: 1.05, 1,10) through a Monte Carlo simulation to create sampling distributions for the average grid-connected emission intensity for each fuel type in Australia. The sampling distributions were then used to determine the best parametric distribution through maximum likelihood fit. The results are shown in Table 4 and Figure 5.

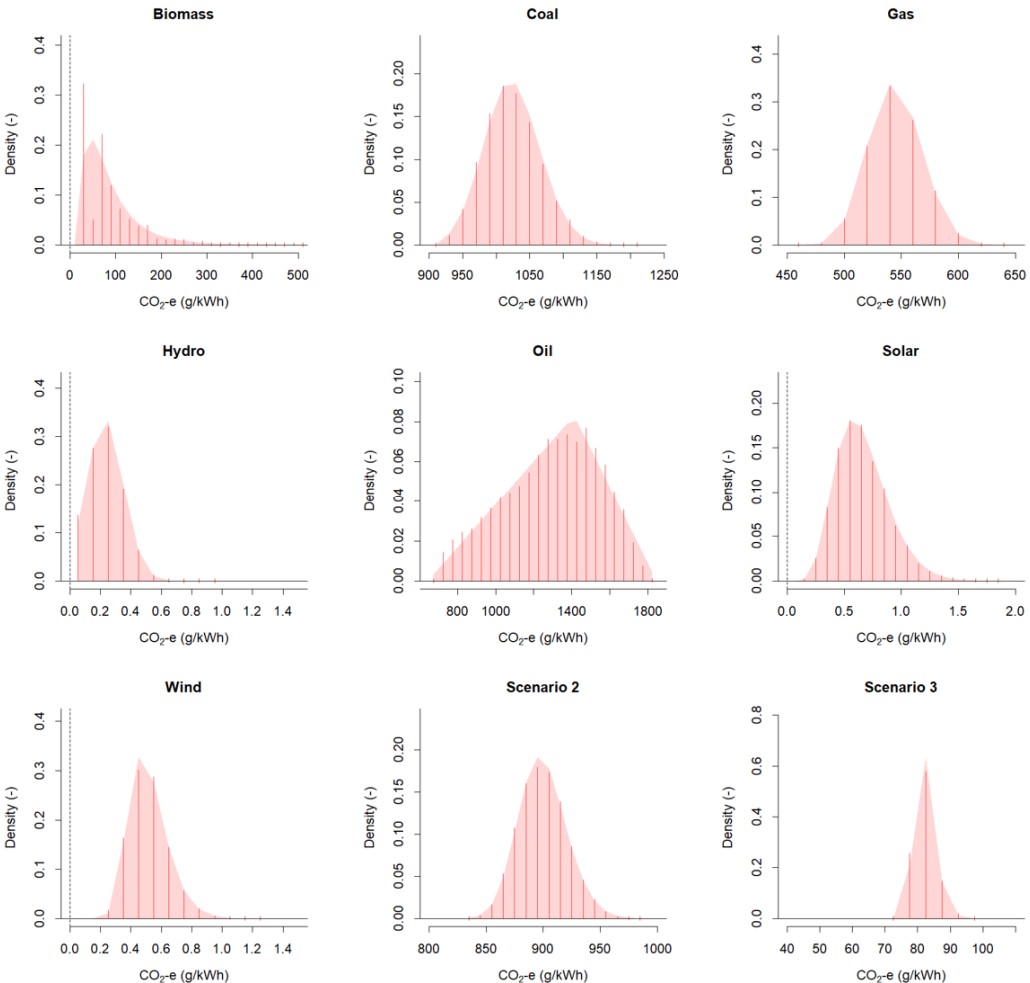

**Figure 5.** GHG emission intensity distributions for grid loss-corrected electricity generation by fuel type (blue) in Australia and for Scenarios 2 and 3 (green), sampling distribution = vertical lines; fitted parametric distribution = shaded polygon.

**Table 4.** GHG emission intensities (g $CO_2$-e/kWh consumed) distribution definitions for grid-loss corrected electricity generation by fuel type in Australia.

| Fuel Type | Distribution | Typical Value | Plausible Min–Max Value |
|---|---|---|---|
| Biomass | Lognormal, **L** (4.27, 0.67) | 89.00 | 27.00–295.00 |
| Coal | Non-standard beta, **B** (7.69, 17.85) | 1023.00 | 913.00–1201.00 |
| Gas | Lognormal, **L** (6.30, 0.04) | 545.00 | 467.00–635.00 |
| Hydro | Normal, **N** (0.23, 0.12) | 0.23 | 0.00–0.80 |
| Oil | Triangular, **T** (638, 1430, 1824) | 1430.00 | 638.00–1824.00 |
| Solar | Gamma, **G** (8.23, 12.42) | 0.66 | 0.14–1.85 |
| Wind | Lognormal, **L** (−0.69, 0.24) | 0.52 | 0.23–1.25 |
| Scenario 2 | Skewed t, **S** (882.01, 27.39, 1.33, 385.26) | 900.00 | 826.00–983.00 |
| Scenario 3 | Skewed t, **S** (78.66, 4.52, 2.87, 27.08) | 82.00 | 74.00–96.00 |

To determine the emission intensity for Scenarios 2 and 3, the fuel type distributions in Table 4 were combined in a Monte Carlo simulation using the corresponding fuel type proportions (Table 1). Skewed t distributions provide the best maximum likelihood fit to the sampling distributions for Scenarios 2 and 3 (Table 4 and Figure 5).

*3.5. On-Road Driving BEVs*

It has been common practice to use type approval laboratory (NEDC, New European Drive Cycle) based measurements of BEV electricity usage reported by vehicle manufacturers. However, NEDC-based data are known to underestimate on-road electricity usage of BEVs by approximately 25 to 35% [5]. Fleet average energy consumption for Australian BEVs is estimated to be 0.19 kWh/km in real-world driving conditions [62]. This value is in line with other studies, which have reported real-world electricity consumption of 0.15 to 0.21 kWh/km for individual BEVs of different weights and sizes [45,47,49]. A plausible range for mean BEV real-world energy consumption in Australian conditions is therefore 0.18–0.21 kWh/km, with a typical value of 0.19 kWh/km. A triangular distribution is assumed for average BEV real-world energy consumption (**T**: 0.18, 0.19, 0.21).

Used electricity by BEVs needs to account for energy losses due to battery recharging. A plausible range for battery charging losses is 5–27%, with a typical value of approximately 15% [5,60,63–67]. Efficiency is computed as 100% minus loss (%) and a triangular distribution is assumed for battery charging efficiency (**T**: 0.73, 0.85, 0.95).

Combining this information with input distributions for electricity generation emission intensity (Section 3.4) in a Monte Carlo simulation estimates an on-road GHG emission factor for BEVs in Australia (Scenario 1) of 175 g $CO_2$-e/km (95% confidence interval = 155–198). For the alternative scenarios with a different fuel/energy mix, the normalized values are 207 g $CO_2$-e/km (95% CI = 183–236) for Scenario 2 and 19 g $CO_2$-e/km (95% CI = 17–22) for Scenario 3. For the Australian jurisdictions, the emission factors are NSW: 182 g $CO_2$-e/km (95% CI = 156–210), VIC: 221 g $CO_2$-e/km (95% CI = 191–256), QLD: 184 g $CO_2$-e/km (95% CI = 159–213), SA: 69 g $CO_2$-e/km (95% CI = 59–80), WA: 154 g $CO_2$-e/km (95% CI = 133–179), TAS: 37 g $CO_2$-e/km (95% CI = 32–43) and NT: 131 g $CO_2$-e/km (95% CI = 113–151). There are large differences in greenhouse gas emission rates from BEVs with Victoria being the highest with 221 g $CO_2$-e/km and Tasmania the lowest with 37 g $CO_2$-e/km, a factor of six difference. This reflects the different electricity generation systems in use in Australia with Victoria mainly relying on brown coal and Tasmania using mainly using hydro power (Table 1). The sampling distributions were used to determine the best theoretical distribution through maximum likelihood fit. The results are shown in Table 5 and Figure 6.

**Table 5.** On-road GHG emission factor (g CO$_2$-e/km) distribution definitions for Australian BEVs by scenario or jurisdiction.

| Fuel Type | Distribution | Typical Value | Plausible Min–Max Value |
|---|---|---|---|
| Scenario 1 (Australia 2018/19) | Non-standard beta, **B** (6.16, 12.20) | 175 | 144–219 |
| Scenario 2 (Marginal Electricity) | Non-standard beta, **B** (5.86, 10.53) | 207 | 170–258 |
| Scenario 3 (More Renewable Electricity) | Lognormal, **L** (2.94, 0.07) | 19 | 14–25 |
| NSW | Non-standard beta, **B** (8.43, 17.03) | 182 | 142–230 |
| VIC | Non-standard beta, **B** (10.88, 24.20) | 221 | 172–284 |
| QLD | Non-standard beta, **B** (8.61, 16.72) | 184 | 144–232 |
| SA | Non-standard beta, **B** (9.70, 20.15) | 69 | 53–90 |
| WA | Non-standard beta, **B** (8.63, 17.20) | 154 | 121–204 |
| TAS | Non-standard beta, **B** (8.80, 19.70) | 37 | 29–48 |
| NT | Non-standard beta, **B** (8.29, 16.14) | 131 | 103–172 |

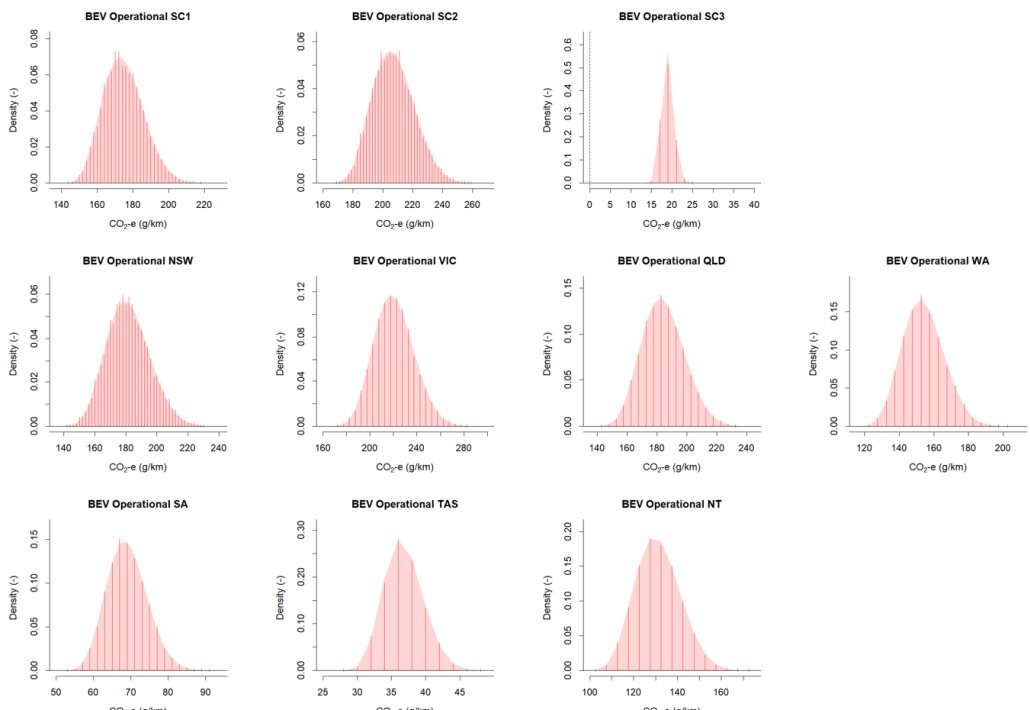

**Figure 6.** On-road GHG emission factor (g CO$_2$-e/km) distributions for Australian BEVs by scenario or jurisdiction. Sampling distribution = vertical lines; fitted parametric distribution = shaded polygon.

### 3.6. Infrastructure for Electricity Generation

Commissioning and decommissioning of fossil-fueled power plants, fossil fuel processing facilities (refineries, fuel storage) and renewable energy sources (wind farms, solar plants, hydro power, etc.) cost energy and generate GHG emissions. Infrastructure GHG emissions are particularly relevant for renewable energy sources. Raw data from a comprehensive LCA review of 33 LCA studies [3] were requested, kindly provided and used to estimate the plausible ranges for infrastructure related GHG intensity per kWh of electricity (generated) by fuel type. The LCA data were used to determine the best parametric distribution through a maximum likelihood fit. When the sample size was too small or when a satisfactory fit could not be obtained, a uniform distribution was assumed. The results are shown in Table 6.

**Table 6.** GHG emission intensities (g $CO_2$-e/kWh generated) distribution definitions for commissioning and decommissioning electricity generation infrastructure by fuel type.

| Fuel Type | Distribution | Typical Value | Plausible Min–Max Value |
|---|---|---|---|
| Biomass | Uniform, **U** (0.04, 2.00) | 0.45 | 0.04–2.00 |
| Coal | Uniform, **U** (0.8, 46.0) | 8.00 | 0.80–46.00 |
| Gas | Triangular, **T** (0.60, 1.85, 3.10) | 1.85 | 0.60–3.10 |
| Hydro | Uniform, **U** (3.10, 20.00) | 7.40 | 3.10–20.00 |
| Oil | Triangular, **T** (1.00, 2.20, 3.00) | 2.20 | 1.00–3.00 |
| Solar | Exponential, **E** (0.015) | 67.94 | 20.00–190.00 |
| Wind | Uniform, **U** (3.00, 41.00) | 18.93 | 3.00–41.00 |

The fuel type distributions in Table 6 were combined in a Monte Carlo simulation with the distributions that were discussed earlier for grid losses (**U**: 1.05, 1,10), BEV real-world energy consumption (**T**: 0.18, 0.21, 0.19) and battery charging efficiency (**T**: 0.73, 0.95, 0.85), using the fuel type percentages as weights (Table 1). The sampling distributions were used to determine the best parametric distribution through maximum likelihood fit. The results are shown in Table 7 and Figure 7.

**Table 7.** Infrastructure GHG emission factor (g $CO_2$-e/km) distribution definitions for Australian BEVs by scenario or jurisdiction.

| Fuel Type | Distribution | Typical Value | Plausible Min–Max Value |
|---|---|---|---|
| Scenario 1 (Australia 2018/19) | Non-standard beta, **B** (2.26, 3.00) | 5.07 | 0.74–10.76 |
| Scenario 2 (Marginal Electricity) | Normal, **N** (4.35, 2.38) | 4.35 | 0.21–9.77 |
| Scenario 3 (More Renewable Electricity) | Lognormal, **L** (1.99, 0.41) | 7.93 | 2.00–19.72 |
| NSW | Non-standard beta, **B** (1.97, 2.77) | 6.07 | 0.66–13.60 |
| VIC | Non-standard beta, **B** (2.33, 3.32) | 5.73 | 0.65–12.92 |
| QLD | Non-standard beta, **B** (1.95, 2.73) | 5.66 | 0.61–12.66 |
| WA | Non-standard beta, **B** (3.05, 4.41) | 2.61 | 0.52–5.59 |
| SA | Non-standard beta, **B** (2.70, 5.98) | 4.35 | 1.00–10.47 |
| TAS | Non-standard beta, **B** (2.31, 2.88) | 3.18 | 0.83–6.11 |
| NT | Gamma, **G** (8.44, 7.95) | 1.06 | 0.38–2.39 |

*3.7. Infrastructure for Fossil Fuels*

For oil refineries, no information could be found to derive an infrastructure GHG emission factor distribution, and the distribution is defined using an alternative approach. It is assumed that GHG emissions due to commissioning and decommissioning of a refinery is similar to an oil-fueled power generation facility. The plausible range then lies between 1.0 and 3.0 g $CO_2$-e/kWh electricity generated (Table 6). The energy content of crude oil is taken as 45.3 MJ/kg fuel [58], which equates to 12.6 kWh/kg fuel. Power plant efficiency is expected to be between 38 and 48%. Combining this information produces an estimate for fossil fuel infrastructure of 5 to 18 g $CO_2$-e per kg of fuel produced. To account for additional uncertainty the plausible range is extended to 2 to 30 g $CO_2$-e per kg of fuel produced. Using the average on-road fuel consumption of 80 g per km for PVs (refer to Section 3.3) then computes an average GHG emission intensity range for refinery infrastructure of approximately 0.2–2.4 g $CO_2$-e per km. A uniform distribution was assumed for $e_{infra,ICEV}$ (**U**: 0.2, 2.5). This range is uncertain, but the error of omission (assuming zero emissions intensity) is considered to be larger than the error in the estimated range.

*3.8. Upstream Emissions for Fossil Fuels*

Extraction, transport, production and distribution of refined fossil fuels such as petrol and diesel require energy and produce GHG emissions. Published well-to-wheel data in the international literature suggest that up to 14 to 28% of the contained energy in the fuels is consumed within the chain, with an estimated average value of 20% [1,3–5,49,50,61]. Combining this range with predicted average fuel consumption for the Australian PV

fleet (Section 3.3), results in an estimate of 35.9 to 72.0 g $CO_2$-e/km, and a typical value of 51.4 g $CO_2$-e/km. A uniform distribution was assumed for $e_{upstream,ICEV}$ (**U**: 35.9, 72.0).

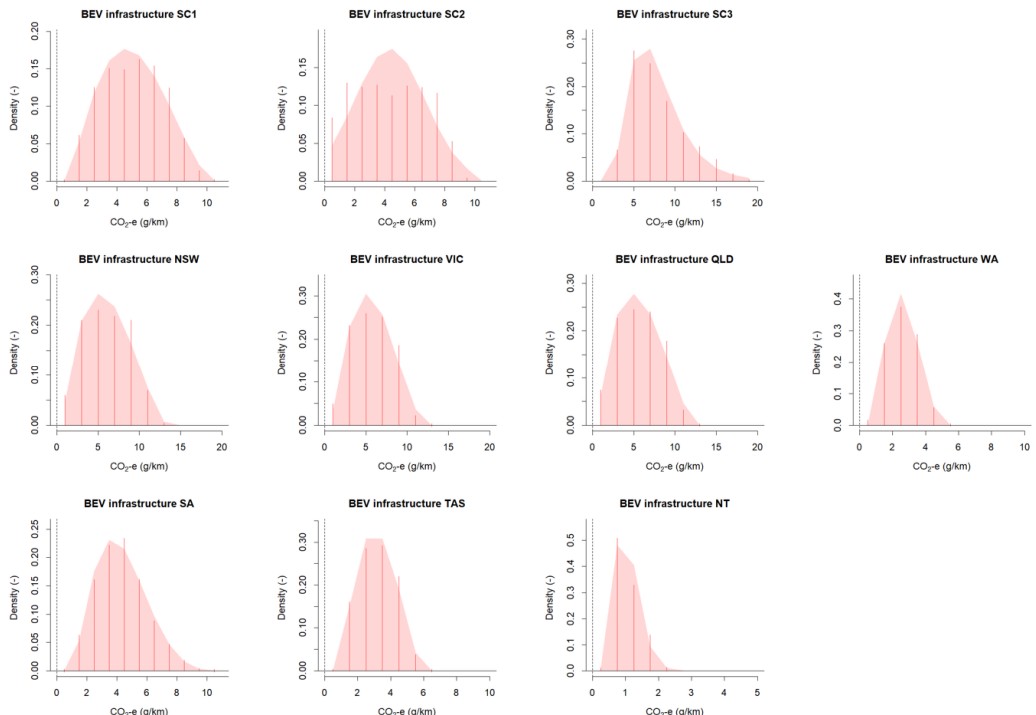

**Figure 7.** Infrastructure GHG emission factor (g $CO_2$-e/km) distributions for Australian BEVs by scenario or jurisdiction. Sampling distribution = vertical lines; fitted parametric distribution = shaded polygon.

### 3.9. Upstream Emissions for Electricity Generation

Upstream emissions for electricity generation are GHG emissions due to upstream extraction, transport, production and distribution of the fossil fuels used in electricity generation. The National Greenhouse Accounts (NGA) provide 'Scope 3' GHG emission factors for electricity production in Australia [58]. Scope 3 accounts for extraction and production of purchased materials and transport of purchased fuels. For Australia, the average upstream GHG emission intensity for consumed electricity is reported as 80 g $CO_2$-e/kWh for the 2018–2019 financial year. This is 10% of the combined Scope 2 and 3 emission intensity.

Upstream emission rates vary greatly among jurisdictions with a minimum of 10 g $CO_2$-e/kWh (WA) and a maximum of 120 g $CO_2$-e/kWh (Queensland). A similar range has been reported in other studies. For instance, upstream emissions for different subregion grids in the USA vary between 27 and 140 g $CO_2$-e/kWh [45]. Given the complexity in quantifying upstream emission factors, the uncertainty in Scope 3 NGA emission factors is expected to be larger than the uncertainty in Scope 2 NGA emission factors for electricity production (5–13%, refer to Section 3.4), which is based on reported fuel use data.

A bootstrap analysis using the upstream USA data (n = 27) and Australian upstream data (n = 7) suggests that the uncertainty (95% CI) in the average upstream emission intensity is approximately ±15% and ±40%, respectively.

The bootstrap sampling distributions were presented as mean bootstrap values divided by the grand mean and subsequently used to determine the best parametric uncertainty distribution through maximum likelihood fit. The Australian data suggest a truncated (min = 0.20, max = 1.80) normal distribution (N: 1.00, 0.22) for the Australian data and a truncated (min = 0.75, max = 1.30) skewed t distribution (S: 0.94, 0.09, 1.49, 216.93) for the USA data. The distributions are shown in Figure 8.

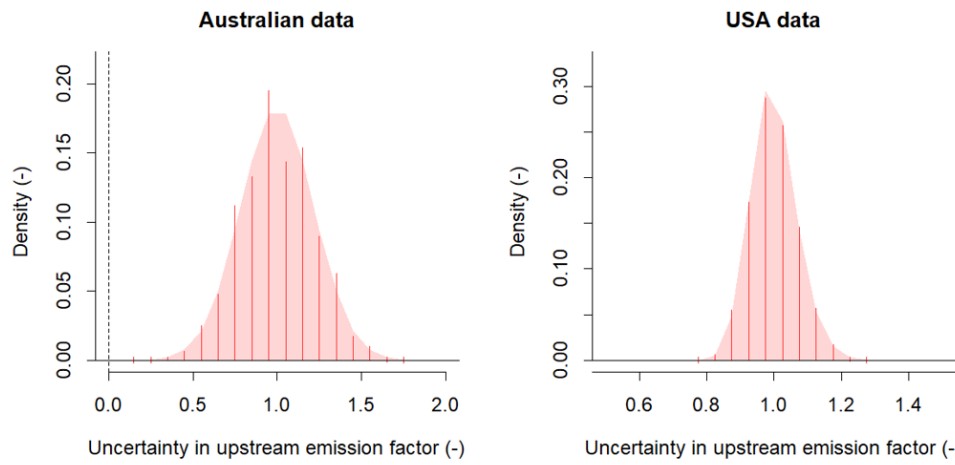

**Figure 8.** Upstream GHG emission factor uncertainty distributions based on Australian and USA upstream data, bootstrap sampling distribution = vertical lines; fitted parametric distribution = shaded polygon.

Given the small sample size of the Australian data, the uncertainty distribution based on the USA data is considered to be more reliable and realistic. The skewed t distribution is therefore used to quantify the uncertainty in the NGA Scope 3 GHG emission factors for electricity production in Australia. These NGA Scope 3 uncertainty distributions were combined in a Monte Carlo simulation with the distributions for battery charging efficiency (**T**: 0.73, 0.95, 0.85) and BEV real-world energy consumption (**T**: 0.18, 0.19, 0.21). The sampling distributions were used to determine the best parametric distribution through maximum likelihood fit. The results are shown in Table 8 and Figure 9.

**Table 8.** Upstream GHG emission factor (g $CO_2$-e/km) distribution definitions for Australian BEVs by jurisdiction based on NGA data.

| Fuel Type | Distribution | Typical Value | Plausible Min–Max Value |
|---|---|---|---|
| Scenario 1 (Australia 2018/19) | Non-standard beta, **B** (14.19, 53.81) | 18.42 | 12.80–26.00 |
| Scenario 2 (Marginal Electricity) | - | - | - |
| Scenario 3 (More Renewable Electricity) | - | - | - |
| NSW | Lognormal, **L** (2.91, 0.09) | 18.39 | 13.00–26.00 |
| VIC | Skewed t, **S** (21.04, 2.86, 1.62, 233.11) | 22.99 | 16.00–32.00 |
| QLD | Skewed t, **S** (25.19, 3.48, 1.67, 102603.40) | 27.57 | 20.00–38.00 |
| WA | Lognormal, **L** (0.83, 0.09) | 2.30 | 1.70–3.30 |
| SA | Lognormal, **L** (2.77, 0.09) | 16.10 | 11.75–22.75 |
| TAS | Skewed t, **S** (4.21, 0.58, 1.64, 2061567.00) | 4.61 | 3.30–6.60 |
| NT | Non-standard beta, **B** (14.35, 40.67) | 11.48 | 7.60–16.00 |

An alternative calculation uses data from the previously discussed LCA meta-study [3]. This study also reported GHG emission intensities for "fuel provision from the extraction of fuel to the gate of the plant". These LCA data were used to determine the best parametric distribution through a maximum likelihood fit. The results are shown in Table 9. It is noted that renewable generation of electricity has an upstream GHG emission intensity of zero (**D**: 0).

As in previous steps, the fuel type distributions in Table 9 were combined in a Monte Carlo simulation with the distributions for grid losses (**U**: 1.05, 1,10), battery charging efficiency (**T**: 0.73, 0.95, 0.85) and BEV real-world energy consumption (**T**: 0.18, 0.21, 0.19), using the fuel type percentages as weights (Table 1). The sampling distributions were used to determine the best parametric distribution through maximum likelihood fit. The results are shown in Table 10 and Figure 10.

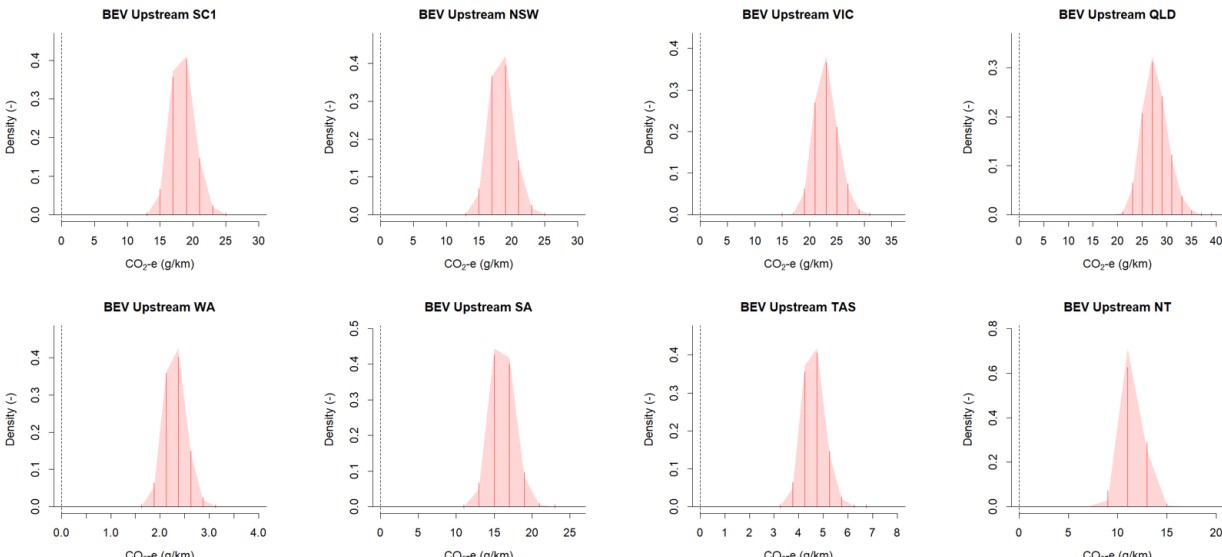

**Figure 9.** Upstream GHG emission factor (g $CO_2$-e/km) distributions for Australian BEVs by scenario or jurisdiction. Sampling distribution = vertical lines; fitted parametric distribution = shaded polygon.

**Table 9.** GHG emission intensities (g $CO_2$-e/kWh generated) distribution definitions for upstream electricity generation infrastructure by fuel type.

| Fuel Type | Distribution | Typical Value | Plausible Min–Max Value |
|---|---|---|---|
| Biomass | Exponential, **E** (0.028) | 35.30 | 1.00–87.00 |
| Coal | Lognormal, **L** (3.81, 0.98) | 66.45 | 7.00–230.00 |
| Gas | Normal, **N** (105.25, 73.29) | 105.25 | 0.56–280.00 |
| Hydro | Dirac, **D** (0.00) | 0.00 | 0.00–0.00 |
| Oil | Gamma, **G** (5.02, 0.20) | 25.60 | 11.00–38.00 |
| Solar | Dirac, **D** (0.00) | 0.00 | 0.00–0.00 |
| Wind | Dirac, **D** (0.00) | 0.00 | 0.00–0.00 |

**Table 10.** Upstream GHG emission factor (g $CO_2$-e/km) distribution definitions for Australian BEVs by scenario or jurisdiction.

| Fuel Type | Distribution | Typical Value | Plausible Min–Max Value |
|---|---|---|---|
| Scenario 1 (Australia 2018/19) | Lognormal, **L** (2.53, 0.53) | 14.18 | 1.00–49.00 |
| Scenario 2 (Marginal Electricity) | Lognormal, **L** (2.73, 0.54) | 17.69 | 1.00–58.00 |
| Scenario 3 (More Renewable Electricity) | Weibull, **W** (2.65, 2.81) | 2.49 | 0.15–6.50 |
| NSW | Skewed t, **S** (2.50, 11.02, 25.32, 4.46) | 13.04 | 1.50–54.50 |
| VIC | Skewed t, **S** (3.00, 10.25, 15.23, 5.20) | 12.53 | 1.40–46.00 |
| QLD | Lognormal, **L** (2.54, 0.57) | 14.94 | 1.70–52.00 |
| WA | Non-standard beta, **B** (2.53, 4.51) | 21.28 | 1.00–58.00 |
| SA | Non-standard beta, **B** (2.03, 3.72) | 13.81 | 0.15–40.00 |
| TAS | Weibull, **W** (1.90, 1.68) | 1.49 | 0.02–4.40 |
| NT | Non-standard beta, **B** (2.06, 3.61) | 23.34 | 0.70–62.50 |

When the typical values in Tables 8 and 10 are compared, it is clear that the two methods generate substantially different results for the jurisdictions, but that the order of magnitude is similar. The NGA based method generates typical values between 2 and 28 g $CO_2$-e/km, whereas the LCA meta-study method generates slightly lower typical values between 1 and 23 g $CO_2$-e/km. The biggest difference is observed for Western Australia (WA) where the NGA Scope 3 factor has a low value of approximately 2 g $CO_2$-e/km but it is unclear why this should be so, given the largely fossil-fueled fuel mix used in this state.

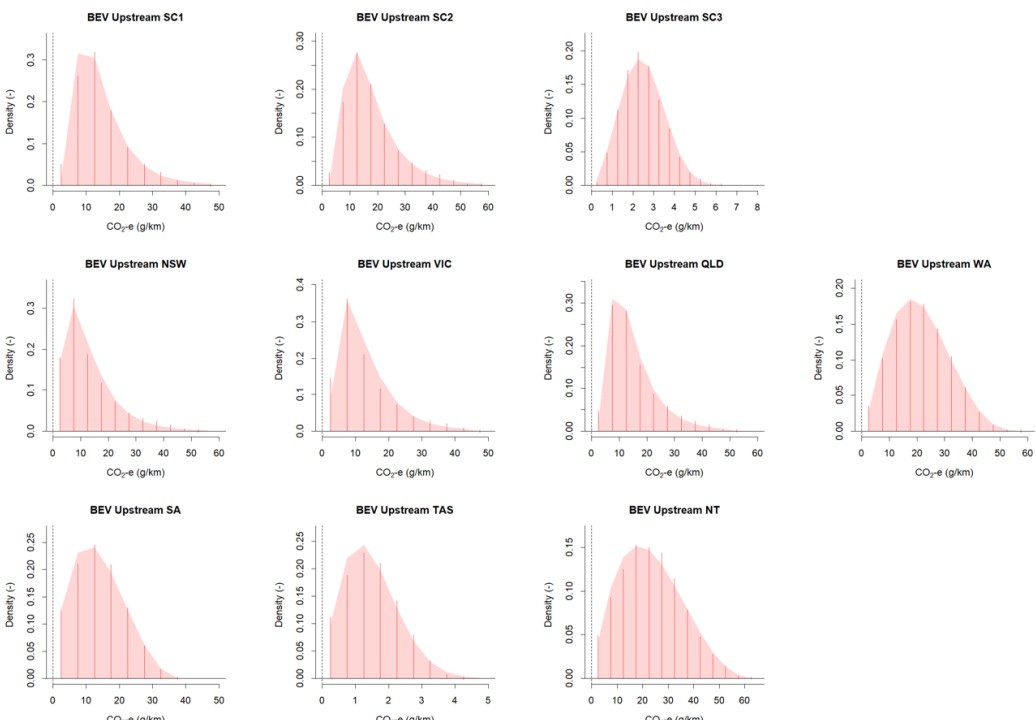

**Figure 10.** Upstream GHG emission factor (g $CO_2$-e/km) distributions for Australian BEVs by scenario or jurisdiction. Sampling distribution = vertical lines; fitted parametric distribution = shaded polygon.

The upstream GHG emission factor distribution definitions in Table 10 appear to generate a more consistent picture. In addition, the LCA meta-study method enables prediction of these emission factors for Scenarios 2 and 3 for which the NGA does not provide information. The results from the LCA meta-study method (Table 10) will therefore be used in the probabilistic technology assessment. The results from the NGA based method (Table 8) will be used to test the sensitivity of the study outcomes. It has conservatively been assumed that the proportion of BEV users that generate their own sustainable electricity (solar panels) for battery recharging is zero.

*3.10. Vehicle Recycling and Disposal*

The assessment of recycling and disposal impacts in an LCA can be prohibitively difficult for a product as complex as a vehicle. GHG emissions from vehicle end-of-life have been found to be small as compared to the operational use phase and are therefore often ignored or included in the vehicle manufacturing LCA aspect [1,68,69]. Moreover, the end-of-life material recycling process of vehicles and batteries will (partly) offset emissions during manufacturing process [21]. Generally, a vehicle's end-of-life impact (recycling and disposal) has a limited contribution in terms of environmental impacts [70]. The impact is dependent on the extent of recycling of vehicle materials.

The Australian Fleet Model shows that approximately 14 million passenger vehicles were active in the Australian on-road fleet in 2018 and that approximately 96% survives each subsequent base year. So, approximately 4% of the vehicles are scrapped, which equates to 560,000 vehicles. The average weight of an Australian PV is 1800 kg, which means that approximately 1 million metric tons of vehicles are scrapped and recycled each year in Australia. A general energy consumption of 66 kWh/ton has been assumed for the recycling process [70].

Using this value in combination with a grid-connected average emission intensity of 760 g $CO_2$-e/kWh (Section 3.4) and the upstream GHG emission intensity of 80 g $CO_2$-e/kWh (Section 3.8), a total of 840 g $CO_2$-e/kWh, estimates 55,884 ton of $CO_2$-e emissions each year due to vehicle recycling. Dividing this value by total VKT (560,000 vehicles times lifetime mileage of 200,000 km), results in a GHG emission rates due to disposal of 0.5 g $CO_2$-e/km.

A plausible range of 0.1 to 2.0 g $CO_2$-e/km has been assumed for vehicle recycling and disposal. The same value is used for ICEVs and BEVs and a uniform distribution was assumed for $e_{disposal,ICEV}$ and $e_{disposal,BEV}$ (**U**: 0.1, 2.0).

It is possible that recycling and disposal of BEVs have a higher GHG impact than ICEVs due to batteries. However, BEVs also have a lighter weight than Australian ICEVs, reducing the impact. In addition, BEV batteries can have a second use as stationary applications to act as a storage buffer for fluctuating renewable electricity generation. This can decrease the vehicle's carbon footprint caused by the battery by 50% [5]. The recovering and recycling of the materials used in BEV batteries has increased significantly due to the high costs of the raw materials for their production [70]. In any case, the differences between BEVs and ICEVs regarding end-of-life recycling or disposal processes are considered to be trivial due to the relatively small impact of this LCA aspect on overall GHG emissions.

## 4. Results and Discussion

### 4.1. Probabilistic Technology Assessment

The parametric distributions for the five life-cycle aspects and two vehicle types were developed in Section 3 (Tables 2, 3, 5, 7 and 10). The reasoning for the MC simulations was slightly different for the full probabilistic LCA, because some life-cycle aspects contain common inputs. Where this was the case ($e_{infra,BEV}$, $e_{upstream,BEV}$, $e_{road,BEV}$, refer to Section 2.2), a lumped emission factor distribution was developed for the three life-cycle aspects combined (Appendix B, Table A2). In the development of the lumped emission factor distributions, the common inputs were first drawn from elicited distributions in each MC simulation and then the life-cycle aspects were computed and then summed, with a new parametric distribution fitted for their sum. These parametric input distributions along with the appropriate parametric distributions fitted in Section 3 were combined in ten separate Monte Carlo simulations with a million simulations for the three Scenarios and seven jurisdictions. The results of the probabilistic technology assessment for the ten simulations are shown in Table 11.

**Table 11.** LCA GHG emission factors (g $CO_2$-e/km) for ICEVs and BEVs by scenario or jurisdiction, including associated uncertainty (95% confidence interval), percent change, probability that any BEV simulation exceeds the minimum value for ICEVs and overlap of confidence intervals.

| Scenario/ Jurisdiction | LCA GHG ICEV g $CO_2$-e/km (95% CI) | LCA GHG BEV g $CO_2$-e/km (95% CI) | Relative Difference % (95% CI) | Probability BEV > ICEV |
|---|---|---|---|---|
| Scenario 1 (Australia Current) | 369 (349 to 390) | 237 (221 to 255) | −36 (−41 to −29) | 0.0 * |
| Scenario 2 (Marginal Electricity) | 369 (349 to 390) | 289 (256 to 328) | −22 (−32 to −10) | $3.6 \times 10^{-4}$ * |
| Scenario 3 (More Renewable Electricity) | 369 (349 to 390) | 85 (74 to 96) | −77 (−80 to −74) | 0.0 * |
| NSW | 368 (344 to 393) | 261 (227 to 301) | −29 (−39 to −17) | $3.0 \times 10^{-6}$ * |
| VIC | 364 (340 to 389) | 287 (257 to 325) | −21 (−31 to −9) | $5.4 \times 10^{-4}$ * |
| QLD | 375 (351 to 400) | 256 (226 to 288) | −32 (−41 to −22) | 0.0 * |
| WA | 387 (363 to 412) | 231 (209 to 255) | −40 (−47 to −33) | 0.0 * |
| SA | 364 (340 to 389) | 143 (126 to 161) | −61 (−66 to −55) | 0.0 * |
| TAS | 369 (343 to 395) | 98 (87 to 109) | −74 (−77 to −70) | 0.0 * |
| NT | 390 (357 to 423) | 218 (194 to 246) | −44 (−52 to −35) | 0.0 * |

* Upper 95% confidence limit BEV simulations < lower 95% confidence limit ICEV simulations.

Explicitly accounting for variability and uncertainty in fleet average GHG emission factors in all relevant life-cycle aspects of ICEVs and BEVs, and the results suggest that, on average, electric passenger vehicles are expected to significantly reduce average life-cycle GHG emission rates for passenger vehicles across all scenarios and all jurisdictions, but that the extent of the reduction in GHG emissions and associated uncertainty varies.

For the current (2018/19) Australian electricity mix (Scenario 1), which is still largely fossil fuels based, the weight of evidence suggests that BEVs will reduce GHG emission rates by 36%, on average, and between 29% and 41%. The last column shows that the

probability that BEVs exceeds the minimum predicted ICEV LCA GHG emission factor is zero, which means that none of the million simulations generated a higher emission rate for BEVs as compared with ICEVs. This suggests that it is highly unlikely that electrification will lead to an adverse policy outcome: an increase in LCA GHG emissions at the fleet level.

For the worst-case 'fossil fuels only' marginal electricity scenario (Scenario 2), the probabilistic analysis shows that electric passenger vehicles are still expected to significantly reduce average GHG emission rates for passenger vehicles. The weight of evidence suggests that, on average, BEVs will reduce GHG emission rates by between 10% and 32% (mean reduction: 22%) for a 100% fossil-fueled marginal electricity mix. There is a small chance of 0.04% that the average BEVs will create equal or higher LCA emissions per kilometer of driving when compared with ICEVs. However, the confidence intervals do not overlap as indicated with the * symbol in the last column.

As of 2018, Australia used more fossil fuels than many other countries such as the EU, USA, Canada, Japan, India, China, South Korea, Russia and Brazil [44]. This is despite the huge potential for renewables in Australia and associated economic and security benefits. Australia may well become a renewable superpower with sufficient political will and support [71]. A more renewable Australian electricity grid mix (Scenario 3) has a substantially lower emissions intensity than the current largely fossil fuel-based grid mix. The probabilistic analysis predicts that electric passenger vehicles are expected to provide large reductions in average LCA GHG emission rates for passenger vehicles. The weight of evidence suggests that the average battery electric vehicle will reduce GHG emission rates by between 74% and 80% (mean reduction: 77%). In the entire MC analysis, there was no simulated BEV mean emission factor which was above the minimum simulated ICEV mean emission factor.

For the Australian jurisdictions, the fleet average LCA GHG emission factors vary substantially for ICEVs (364–390 g $CO_2$-e/km), but particularly for BEVs (98–287 g $CO_2$-e/km), which reflects the differences in fuel mix for electricity generation in the different states and territories (Table 1). As a consequence, the potential reduction in LCA GHG emissions per vehicle kilometer through electrification of the on-road fleet is different, as is shown in Figure 11. Figure 11 presents box plots for each scenario or jurisdiction showing the results of the probabilistic analysis for the absolute and relative differences in fleet average GHG emission rate distributions of BEVs and ICEVs.

Electrification of the Tasmania (TAS) on-road fleet has the largest emission reduction with a predicted reduction of 272 g $CO_2$-e/km (243–300 g $CO_2$-e/km), closely followed by South Australia (SA) with 222 g $CO_2$-e/km (191–252 g $CO_2$-e/km). It demonstrates that electrification in two Australian states will already achieve large reductions in GHG emissions from passenger vehicles of 74% (TAS) and 61% (SA), respectively.

At the other end of the spectrum, the smallest absolute reduction, although still significant, is predicted for Victoria (VIC) and New South Wales (NSW), with 77 (32–117 g $CO_2$-e/km) and 108 g $CO_2$-e/km (60–151 g $CO_2$-e/km), respectively, for the current situation. Electrification in these two Australian states is expected to achieve significant reductions in 2018/2019 in GHG emissions from passenger vehicles of 21% (VIC) and 29% (NSW), respectively. However, these values will improve substantially as the electricity generation system is further decarbonized. This is evident from the positive results that were obtained for Scenario 3 (more renewables), Tasmania and South Australia.

There is a negligible chance of 0.05% in Victoria and 0.0003% in New South Wales that BEVs will create equal or higher LCA emissions per kilometer of driving when compared with ICEVs. Additionally, the confidence intervals do not overlap, indicating that this result is not plausible and meaning that electrification are expected to generate greenhouse emission reduction benefits. For the other jurisdictions, Scenarios 1 and 3, none of the million simulations, spanning the full range of possible outcomes, resulted in BEVs having higher LCA emissions per kilometer of driving when compared with ICEVs, and the computed probability of this occurring is therefore zero.

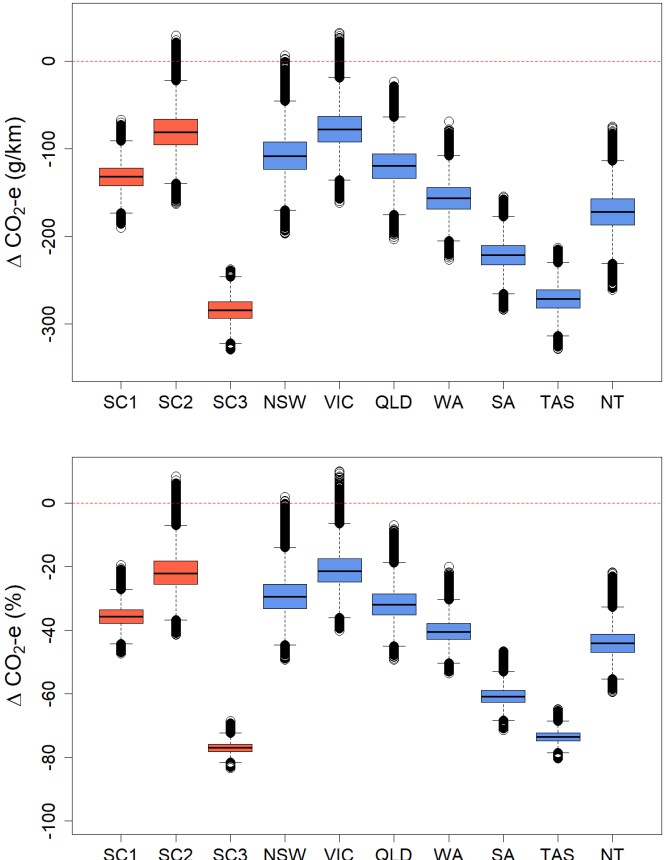

**Figure 11.** Box plots of the absolute (**top**) and relative (**bottom**) differences in LCA GHG emission rates between BEVs and ICEVs by scenario (red) or jurisdiction (blue).

Despite the unique characteristics of the Australian on-road fleet, the results from this study appear to align well with general findings published in the international scientific literature [8,28]. For instance, a review of 51 environmental life-cycle assessments [8] concluded that BEVs powered by coal-fired electricity appear to perform better than conventional ICEVs in terms of GWP.

### 4.2. Sensitivity Analysis

Alternative parametric distributions were developed for (a) the on-road ICEVs and (b) upstream BEV LCA aspects.

(a) Using COPERT Australia and the Australian Fleet Model, a non-standard beta distribution (**B**: 9.89, 16.86) with truncation at 225 and 298 g $CO_2$-e/km was developed for on-road ICEVs (Section 3.3). The alternative parametric distribution was used in a repeat of the probabilistic technology assessment for the current (2018/19) Australian electricity mix (Scenario 1). The results are presented in Appendix C. The mean LCA GHG emission factor for ICEVs is reduced from 369 to 356 g $CO_2$-e/km (3.5%), compared to the probabilistic technology assessment in Section 4.1. The overall predicted effect of electrification (Figure 11) is similar. Section 4.1 predicted that BEVs will reduce GHG emission rates by 36% on average and by between 29% and 41%. Using COPERT Australia and the Australian Fleet Model as an alternative input, it is predicted that BEVs will reduce GHG emission rates by 33% on average and by between 26% and 40%, a similar result. The probability that BEVs exceeds the minimum predicted ICEV LCA GHG emission factor is zero for both simulations, which means that none of the million simulations generated a higher emission rate for BEVs as compared with ICEVs.

(b) Alternative upstream GHG emission factor distributions for Australian BEVs were developed using Scope 3 data from the National Greenhouse Accounts and overseas publications (Section 3.8). The alternative parametric distributions (Table 8) were used in a repeat of the probabilistic technology assessment, including development of alternative lumped emission factor distributions (Appendix B, Table A3). The results are presented in Appendix D. The mean LCA GHG emission factors for BEVs vary by approximately ±15%, compared to the probabilistic technology assessment in Section 4.1. The largest change is observed for NT, where the mean BEV GHG emission factor is reduced by 14% from 218 to 188 g $CO_2$-e/km. For the majority of jurisdictions, this difference is typically approximately ±5% or less. The overall predicted effect of electrification (Figure 11) is remarkably stable as is shown in Figure A2, which suggests that the results from the probabilistic analysis are robust.

### 4.3. Expansion and Refinement

This study has focused on quantifying GHG emissions performance and associated uncertainty for Australian passenger vehicles specifically at the fleet level.

The LCA model can readily be expanded to conduct simulations at a higher level of detail. For instance, the Australian Fleet Model can be used to quantify input distributions for a more disaggregated fleet definition such as small, medium and large passenger cars and compact and large SUVs. An even more detailed analysis specifically considering vehicle make and model is also possible. This expansion would lead to more detailed simulations and can be used to explore various what-if scenarios, for instance by exploring different purchasing behavior scenarios and assessing the impacts on GHG emission performance for both BEVs and ICEVs.

Another possible expansion is to conduct the analysis for trucks and buses, to include other emerging zero emission technologies such as hydrogen vehicles, biofuels and e-fuels and/or to consider other units of assessment such as costs per kilometer (total cost of ownership), air pollutant emissions per km and total life-cycle cost.

Existing LCA software tools such as SimaPro and Open LCA [28] can be readily integrated in the probabilistic LCA framework presented in this paper. They can, for instance, be used to further refine the LCA model definition and subsequently provide the necessary input data for the development of input distributions in the probabilistic LCA.

The pLCA approach presented in this paper is versatile, highly flexible and cost-effective in its application, as well as relatively fast, transparent and intuitive. These features are useful in a time where action to reduce greenhouse gas emissions is urgent and robust and scientifically sound methods to assess a range of technologies are required for cost-effective policy development.

### 5. Conclusions

Life-cycle assessment (LCA) is a powerful and holistic method. However, given the complexity and wide scope of LCAs, it is important to quantify and report variability and uncertainty in the study outcomes. This study uses probabilistic LCA (pLCA) to explicitly model and assess uncertainty in the inputs and results. A technology assessment is conducted for battery electric and conventional fossil-fueled passenger vehicles for three Australian scenarios (current, marginal electricity and more renewables) and seven Australian states and territories. Parametric input distributions were developed by applying statistical techniques to available empirical input data, software output and published data.

The results suggest that electric passenger vehicles are expected to significantly reduce fleet average life-cycle GHG emission rates (g $CO_2$-e/km) for passenger vehicles for all Scenarios and for all jurisdictions, but that the extent of the reduction in GHG emissions and associated uncertainty varies. For the current (2018/19) Australian electricity mix (Scenario 1), which is still largely fossil fuels based, the weight of evidence suggests that BEVs will reduce GHG emission rates by 36% on average (95% confidence interval: 29% to 41%). For the worst-case 'fossil fuels only' marginal electricity scenario (Scenario 2) electric

passenger vehicles are still expected to significantly reduce average GHG emission rates for passenger vehicles between 10% and 32%. Large reductions by between 74% and 80% in fleet average LCA GHG emission rates for passenger vehicles through electrification are predicted for more renewables (Scenario 3).

For the individual Australian states and territories, the fleet average LCA GHG emission factors vary substantially for ICEVs (364–390 g $CO_2$-e/km), but particularly for BEVs (98–287 g $CO_2$-e/km), which reflects the differences in fuel mix for electricity generation in the different states and territories. Electrification of the Tasmania (TAS) on-road fleet has the largest emission reduction with a predicted absolute value of 272 g $CO_2$-e/km (243–300 g $CO_2$-e/km), closely followed by South Australia (SA) with 222 g $CO_2$-e/km (191–252 g $CO_2$-e/km). This demonstrates that electrification in two Australian states will already achieve large reductions in GHG emissions from passenger vehicles of 74% (TAS) and 61% (SA), respectively.

Typically, none of the million simulations resulted in (fleet average) BEVs having higher LCA emissions per kilometer of driving when compared with (fleet average) ICEVs. In a few cases, the study found a negligible probability (equal or less than 0.05%) that BEVs will create equal or higher LCA emissions per kilometer of driving when compared with ICEVs, but generally none of the simulations indicated an adverse policy outcome for electrification. This means that after considering the complete vehicle life cycle, the weight of evidence strongly suggests that rapid electrification of the Australian on-road fleet away from fossil fuels is a safe and effective policy measure to reduce greenhouse gas emissions from road transport. A sensitivity analysis with alternative input distributions demonstrated that the outcomes from this study are robust.

**Author Contributions:** Conceptualization, R.S.; methodology, R.S. and D.W.K.; software, R.S.; validation, R.S.; formal analysis, R.S. and D.W.K.; investigation, R.S.; resources, R.S.; data curation, R.S.; writing—original draft preparation, R.S.; writing—review and editing, R.S. and D.W.K.; visualization, R.S.; supervision, R.S.; project administration, R.S.; funding acquisition, R.S. All authors have read and agreed to the published version of the manuscript.

**Funding:** This research was internally funded by Transport Energy/Emission Research Pty. Ltd., Brisbane, Australia (https://www.transport-e-research.com/, accessed on 12 February 2022).

**Institutional Review Board Statement:** Not applicable.

**Informed Consent Statement:** Not applicable.

**Data Availability Statement:** Not applicable.

**Conflicts of Interest:** The authors declare no conflict of interest.

## Appendix A. Distribution Definitions

**Table A1.** Distribution definitions.

| Name | Range | Parameters | Probability Density Function (PDF) |
|---|---|---|---|
| Uniform—$\mathbf{U}(x{:}a,b)$ | $a \leq x \leq b$ | $a$: Minimum, $-\infty < a < b$ <br> $b$: Maximum, $a < b < -\infty$ | $\frac{1}{b-a}$ |
| Triangular—$\mathbf{T}(x{:}a,b,c)$ | $a \leq x \leq b$ | $a$: Minimum, $-\infty < a < b$ <br> $b$: Maximum, $a < b < -\infty$ <br> $c$: Mode, $a \leq c \leq b$ | $\begin{cases} \frac{2(x-a)}{(b-a)(c-a)}, & x \leq c \\ \frac{2(b-x)}{(b-a)(c-a)}, & x > c \end{cases}$ |
| Normal—$\mathbf{N}(x{:}m,s)$ | $-\infty \leq x \leq +\infty$ | $m$: Mean, $-\infty < m < \infty$ <br> $s$: Standard deviation, $0 < s < \infty$ | $\frac{1}{\sqrt{2\pi}s} \exp\left(-\frac{1}{2s^2}(x-m)^2\right)$ |
| Lognormal—$\mathbf{L}(x{:}m,s)$ | $0 \leq x \leq +\infty$ | $m$: Log-mean, $-\infty < m < \infty$ <br> $s$: Scale, $0 < s < \infty$ | $\frac{1}{x\sqrt{2\pi}s} \exp\left(-\frac{1}{2s^2}(ln(x)-m)^2\right)$ |
| Weibull—$\mathbf{W}(x{:}s,k)$ | $0 \leq x \leq +\infty$ | $s$: Scale, $0 < s < \infty$ <br> $k$: Shape, $0 < s < \infty$ | $\frac{k}{s}\left(\frac{x}{s}\right)^{k-1} \exp\left(-\left(\frac{x}{s}\right)^k\right)$ |

**Table A1.** *Cont.*

| Name | Range | Parameters | Probability Density Function (PDF) |
|---|---|---|---|
| Gamma—**G**(*x*:*s*,*k*) | $0 \leq x \leq +\infty$ | *s*: Scale, $0 < \text{s} < \infty$ <br> *r*: Rate, $0 < \text{s} < \infty$ | $\frac{r^s}{\Gamma(s)}x^{s-1}\exp(-rx)$ |
| Exponential—**E**(*x*:*s*) | $0 \leq x \leq +\infty$ | *s*: Scale, $0 < \text{s} < \infty$ | $r\exp(-rx)$ |
| Non-Standard Beta—**B**(*x*:*s*,*k*,*a*,*b*) | $a \leq x \leq b$ | *s*: Scale, $0 < \text{s} < \infty$ <br> *k*: Shape, $0 < \text{k} < \infty$ <br> *a*: Minimum, $-\infty < a < b$ <br> *b*: Maximum, $a < b < -\infty$ | $\frac{\Gamma(s+k)}{\Gamma(s)\Gamma(k)}\left(\frac{x-a}{b-a}\right)^{s-1}\left(1-\frac{x-a}{b-a}\right)^{k-1}$ |
| Skew t—**S**(*x*:*m*,*s*,*a*,*d*) | $-\infty \leq x \leq +\infty$ | *m*: Mean, $-\infty < m < \infty$ <br> *s*: Scale, $0 < \text{s} < \infty$ <br> *a*: Skew, $0 < \text{a} < \infty$ <br> *d*: Degrees of freedom, $0 < \text{d} < \infty$ | $2t(x:m,s,d)T\left(az\sqrt{\frac{d+1}{d+z^2}}:0,1,d\right)$, where $t(x:m,s,d)=$ $\frac{\Gamma\left(\frac{1}{2}(d+1)\right)}{\frac{\sqrt{\pi d1}}{2}d}\left(1+\left(\frac{x-m}{s}\right)^2\right)^{-\frac{v+1}{2}}$ $z=(x-m)/s$, and $T(x:m,s,d)$ is the cumulative distribution function. * |
| Dirac Delta—**D**(*x*:*m*) | $-\infty \leq x \leq +\infty$ <br> Practically $x = m$ | *m*: Location, $-\infty < m < \infty$ | $\begin{cases}\infty, & x = m \\ 0, & x \neq m\end{cases}$ |

* See [35] for more details.

## Appendix B. Lumped GHG Distributions

**Table A2.** Lumped (infra, upstream, road) GHG emission factor (g $CO_2$-e/km) distribution definitions for Australian BEVs by scenario or jurisdiction.

| Fuel Type | Distribution | Typical Value | Plausible Min–Max Value |
|---|---|---|---|
| Scenario 1 (Australia 2018/19) | Non-standard beta, **B** (8.91, 29.73) | 194.00 | 153.00–257.00 |
| Scenario 2 (Marginal Electricity) | Skewed t, **S** (210.66, 25.20, 1.96, 19870.21) | 229.00 | 177.00–306.00 |
| Scenario 3 (More Renewable Electricity) | Non-standard beta, **B** (5.32, 24.15) | 29.00 | 20.00–46.00 |
| NSW | Skewed t, **S** (182.18, 25.83, 1.91, 19344.81) | 200.00 | 148.00–281.00 |
| VIC | Skewed t, **S** (209.99, 23.74, 2.07, 379.39) | 227.00 | 184.00–300.00 |
| QLD | Lognormal, **L** (5.27, 0.08) | 196.00 | 156.00–263.00 |
| WA | Non-standard beta, **B** (7.26, 16.91) | 177.00 | 136.00–251.00 |
| SA | Non-standard beta, **B** (7.26, 16.91) | 86.00 | 62.00–127.00 |
| TAS | Non-standard beta, **B** (6.53, 11.81) | 38.00 | 31.00–49.00 |
| NT | Non-standard beta, **B** (5.83, 12.95) | 163.00 | 122.00–238.00 |

**Table A3.** Lumped (infra, upstream, road) GHG emission factor (g $CO_2$-e/km) distribution definitions for Australian BEVs by scenario or jurisdiction used in Section 4.2 based on alternative parametric upstream distributions (Table 8).

| Fuel Type | Distribution | Typical Value | Plausible Min–Max Value |
|---|---|---|---|
| Scenario 1 (Australia 2018/19) | Non-standard beta, **B** (5.30, 9.63) | 195.00 | 165.00–242.00 |
| Scenario 2 (Marginal Electricity) | - | - | - |
| Scenario 3 (More Renewable Electricity) | - | - | - |
| NSW | Non-standard beta, **B** (9.37, 16.99) | 206.00 | 161.00–265.00 |
| VIC | Non-standard beta, **B** (5.32, 9.49) | 245.00 | 205.00–297.00 |
| QLD | Non-standard beta, **B** (5.81, 10.54) | 237.00 | 198.00–288.00 |
| WA | Non-standard beta, **B** (4.63, 8.22) | 164.00 | 137.00–201.00 |
| SA | Non-standard beta, **B** (8.32, 14.88) | 90.00 | 75.00–109.00 |
| TAS | Non-standard beta, **B** (7.26, 12.69) | 45.00 | 37.00–55.00 |
| NT | Non-standard beta, **B** (4.59, 8.11) | 130.00 | 111.00–157.00 |

## Appendix C. Sensitivity Analysis Using Alternative On-Road ICEV Distribution

**Table A4.** LCA GHG emission factors (g $CO_2$-e/km) for ICEVs and BEVs by scenario or jurisdiction, including associated uncertainty (95% confidence interval), percent change, probability that any BEV simulation exceeds the minimum value for ICEVs and overlap of confidence intervals.

| Scenario/ Jurisdiction | LCA GHG ICEV g $CO_2$-e/km (95% CI) | LCA GHG BEV g $CO_2$-e/km (95% CI) | Relative Difference % (95% CI) | Probability BEV > ICEV |
|---|---|---|---|---|
| Scenario 1 (Australia Current) | 356 (332 to 381) | 237 (221 to 255) | −33 (−40 to −26) | 0.0 * |

\* Upper 95% confidence limit BEV simulations < lower 95% confidence limit ICEV simulations.

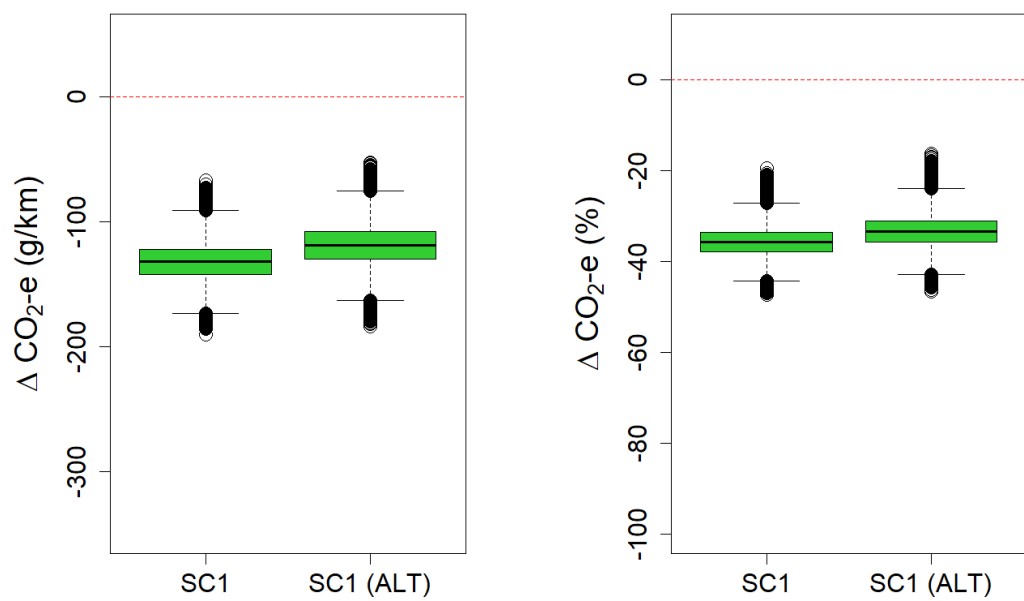

**Figure A1.** Box plots of the absolute (left) and relative (right) differences in LCA GHG emission rates between BEVs and ICEVs for Scenario 1 (Section 4.1) and Alternative Scenario 1 (Section 4.2).

## Appendix D. Sensitivity Analysis Using Alternative Upstream BEV GHG Distributions

**Table A5.** LCA GHG emission factors (g $CO_2$-e/km) for ICEVs and BEVs by scenario or jurisdiction, including associated uncertainty (95% confidence interval), percent change, probability that any BEV simulation exceeds the minimum value for ICEVs and overlap of confidence intervals.

| Scenario/ Jurisdiction | LCA GHG ICEV g $CO_2$-e/km (95% CI) | LCA GHG BEV g $CO_2$-e/km (95% CI) | Relative Difference % (95% CI) | Probability BEV > ICEV |
|---|---|---|---|---|
| Scenario 1 (Australia Current) | 369 (349 to 390) | 250 (231 to 270) | −32 (−25 to −39) | 0.0 * |
| Scenario 2 (Marginal Electricity) | - | - | - | - |
| Scenario 3 (More Renewable Electricity) | - | - | - | - |
| NSW | 368 (344 to 393) | 258 (238 to 280) | −30 (−37 to −22) | 0.0 * |
| VIC | 364 (340 to 389) | 298 (276 to 323) | −18 (−26 to −9) | $6.1 \times 10^{-5}$ * |
| QLD | 375 (351 to 400) | 290 (268 to 314) | −23 (−30 to −14) | 0.0 * |
| WA | 387 (363 to 412) | 220 (202 to 240) | −43 (−49 to −37) | 0.0 * |
| SA | 364 (340 to 389) | 147 (135 to 160) | −60 (−64 to −55) | 0.0 * |
| TAS | 369 (343 to 395) | 104 (93 to 115) | −72 (−75 to −68) | 0.0 * |
| NT | 390 (357 to 423) | 188 (173 to 204) | −52 (−57 to −46) | 0.0 * |

\* Upper 95% confidence limit BEV simulations < lower 95% confidence limit ICEV simulations.

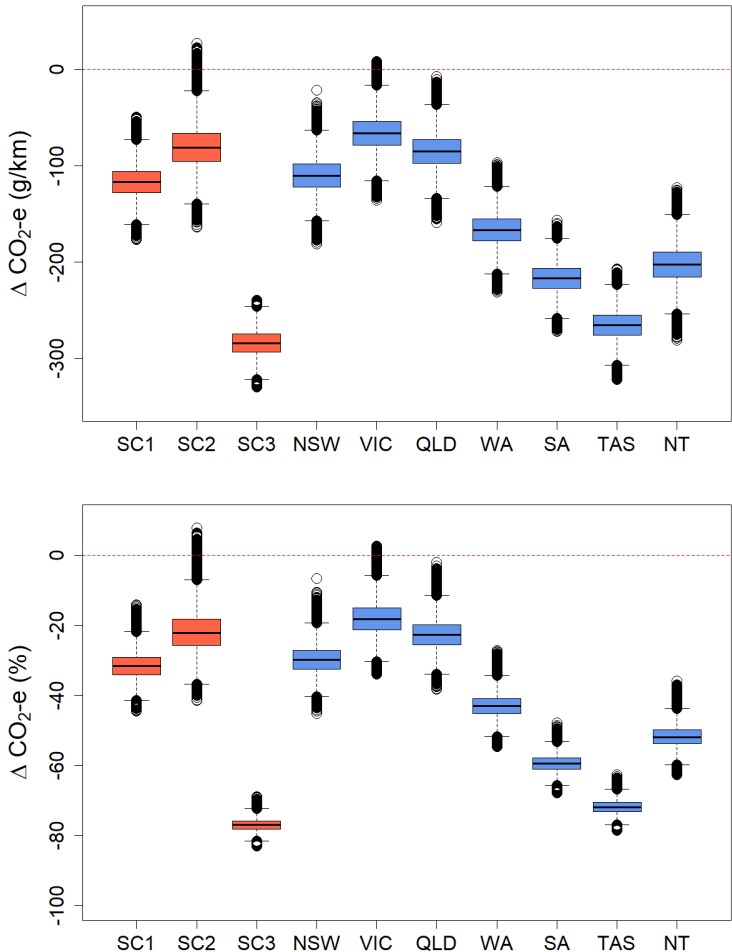

**Figure A2.** Box plots of the absolute (**top**) and relative (**bottom**) differences in LCA GHG emission rates between BEVs and ICEVs by scenario (red) or jurisdiction (blue).

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
