# Peer review of "Greenhouse Gas Emissions Performance of Electric and Fossil-Fueled Passenger Vehicles with Uncertainty Estimates Using a Probabilistic Life-Cycle Assessment"

_sustainability, doi:10.3390/su14063444_

Round 1

Reviewer 1 Report

I think the authors did interesting research, but I think I found a major inconsistency in the fundament of their model.

Based on the authors' description of their model from Line 91 to 98:

The unit in Equation 2(a) is kg CO2 emission;

But the unit in Equation 2(d) is kg CO2 emission per kilometer.

Therefore, there is no way one can combine these two terms of different units into a single equation (Equation(2)).

Similar unit issues are also found in Figure 1, where the carbon emission of car manufacturing is under the unit of g emission per kilometer, while it should be in g emission per kg of the car's weight, according to the statements in Line 92.

I did not finish reading the manuscript after detecting these serious problems, because I do not want to try the apples from a twisted tree.

I think the authors should sit back to their computer and double-check their program.

Author Response

We thank the reviewer for the pickup on units. What was missing from the Equation 2a was lifetime mileage, as was explained in section 3.2. The computations that were made were in fact correct, as discussed in section 3.2, but we have corrected Equation 2a (as well as 1a). We have also changed the overall calculation to include a lumped distribution approach to better account for dependencies between life cycle aspects as discussed in the updated paper.

Reviewer 2 Report

Journal: Sustainability

Comments on the manuscript entitled “Title: Greenhouse gas emissions performance of electric and fossil-fueled passenger vehicles with uncertainty estimates using a probabilistic life-cycle assessment (pLCA)” (Manuscript No. sustainability-1495065)

The article is not suitable for publication in its present form. It needs a major revision. Below are my comments:

Some specific comments:

  1. Abstract: It is suggested to add some background with a few objectives and possible applications of this study and highlight the novelty of this work. The abstract only contains some parameters without any process conditions or key values from results, which is insufficient to delineate the whole pictures of contribution and possible application of this study.
  2. Revise keywords add more specific and novel keywords with broader meanings (5-7 words). Don’t use abbreviations in the keyword.
  3. The introduction is lacking sufficient background information, which is unable to give the reader detailed background knowledge and possible wide application of this study. Research gaps should be highlighted more clearly and future applications of this study should be added.
  4. Compare the emission results with any country results data.
  • Yaqoob H, Teoh YH, Goraya TS, Sher F, Jamil MA, Rashid T, et al. Energy evaluation and environmental impact assessment of transportation fuels in Pakistan. Case Studies in Chemical and Environmental Engineering 2021;3:100081. doi:10.1016/j.cscee.2021.100081.
  • Al-Juboori, F. Sher, U. Khalid, M.B.K. Niazi, G.Z. Chen, Electrochemical

production of sustainable hydrocarbon fuels from CO2Co-electrolysis in eutectic

molten melts, ACS Sustain. Chem. Eng. 8 (2020) 12877–12890,https://doi.org/

10.1021/acssuschemeng.0c03314.

  • Yaqoob H, Teoh YH, Sher F, Ashraf MU, Amjad S, Jamil MA, et al. Jatropha Curcas Biodiesel : A Lucrative Recipe for Pakistan’s Energy Sector. Processes 2021;9:1129.
  • H€ o€ ok, X. Tang, Depletion of fossil fuels and anthropogenic climate change—a review, Energy Pol. 52 (2013) 797–809,https://doi.org/10.1016/j.enpol.2012.10.046.
  • Yaqoob H, Teoh YH, Ud Din Z, Sabah NU, Jamil MA, Mujtaba MA, et al. The potential of sustainable biogas production from biomass waste for power generation in Pakistan. Journal of Cleaner Production 2021;307:127250. doi:https://doi.org/10.1016/j.jclepro.2021.127250.
  1. Compare the results of electric vehicles and fossil fuels with alternative fuels.
  • Yaqoob H, Teoh YH, Sher F, Ashraf MU, Amjad S, Jamil MA, et al. Jatropha Curcas Biodiesel : A Lucrative Recipe for Pakistan’s Energy Sector. Processes 2021;9:1129.
  • Yaqoob H, Teoh YH, Sher F, Farooq MU, Jamil A, Kausar Z, et al. Potential of Waste Cooking Oil Biodiesel as Renewable Fuel in Combustion Engines : A Review. Energies 2021;14:2565. doi:https://doi.org/10.3390/en14092565.
  1. The tables/figures inserted are not explained or discussed well in the text please discuss critically and explain all tables/figures in the text.
  2. Against which standard was the engine used homologated?
  3. What are the tolerances for the emissions equipment? Cheaper multigas analyzers are notorious for having very high uncertainties and are generally not acceptable for academic publications.
  4. There appears to be some sort of smoothing on the pressure data. How was this performed?
  5. No error bars were put in the figures.
  6. Conclusions of this manuscript lacks clear findings and future aspects. The authors are advised to write the conclusion comprehensively.
  7. There are some grammatical issues, particularly in the introduction, and results & discussion section.
  8. Validate the results section with the help of physicochemical properties not just with the previously published results.
  9. Reduce the similarity to below 20%.
  10. Between the lack of methodology and limited novelty, the authors have a significant amount of work to do to bring this paper up to a publishable standard.

Author Response

Some specific comments:

  1. Abstract: It is suggested to add some background with a few objectives and possible applications of this study and highlight the novelty of this work. The abstract only contains some parameters without any process conditions or key values from results, which is insufficient to delineate the whole pictures of contribution and possible application of this study.

Response authors: We have to comply with the journal’s directions for the Abstract: A single paragraph of about 200 words maximum. We note that it was a challenge to achieve this word limit and we had to make strict decisions what to include. There is unfortunately no space left to add more text without removing current text which in our view is important to retain.

  1. Revise keywords add more specific and novel keywords with broader meanings (5-7 words). Don’t use abbreviations in the keyword.

Response authors: We have to comply with the journal’s direction on the matter: List three to ten pertinent keywords specific to the article yet reasonably common within the subject discipline. We believe we have done this but are happy to change if this is not the case.

  1. The introduction is lacking sufficient background information, which is unable to give the reader detailed background knowledge and possible wide application of this study. Research gaps should be highlighted more clearly and future applications of this study should be added.

Response authors: We have expanded and added to the introduction section along the lines suggested and added 15 new references. We also note that section 4.3 already discussed future application and refinement/expansion of the method in case the reviewer overlooked this.

  1. Compare the emission results with any country results data.
  • Yaqoob H, Teoh YH, Goraya TS, Sher F, Jamil MA, Rashid T, et al. Energy evaluation and environmental impact assessment of transportation fuels in Pakistan. Case Studies in Chemical and Environmental Engineering 2021;3:100081. doi:10.1016/j.cscee.2021.100081.
  • Al-Juboori, F. Sher, U. Khalid, M.B.K. Niazi, G.Z. Chen, Electrochemical production of sustainable hydrocarbon fuels from CO2Co-electrolysis in eutectic molten melts, ACS Sustain. Chem. Eng. 8 (2020) 12877–12890,https://doi.org/10.1021/acssuschemeng.0c03314.
  • Yaqoob H, Teoh YH, Sher F, Ashraf MU, Amjad S, Jamil MA, et al. Jatropha Curcas Biodiesel : A Lucrative Recipe for Pakistan’s Energy Sector. Processes 2021;9:1129.
  • H€ o€ ok, X. Tang, Depletion of fossil fuels and anthropogenic climate change—a review, Energy Pol. 52 (2013) 797–809,https://doi.org/10.1016/j.enpol.2012.10.046.
  • Yaqoob H, Teoh YH, Ud Din Z, Sabah NU, Jamil MA, Mujtaba MA, et al. The potential of sustainable biogas production from biomass waste for power generation in Pakistan. Journal of Cleaner Production 2021;307:127250. doi:https://doi.org/10.1016/j.jclepro.2021.127250.

Response authors: This suggestion is outside the scope of this paper.

  1. Compare the results of electric vehicles and fossil fuels with alternative fuels.
  • Yaqoob H, Teoh YH, Sher F, Ashraf MU, Amjad S, Jamil MA, et al. Jatropha Curcas Biodiesel : A Lucrative Recipe for Pakistan’s Energy Sector. Processes 2021;9:1129.
  • Yaqoob H, Teoh YH, Sher F, Farooq MU, Jamil A, Kausar Z, et al. Potential of Waste Cooking Oil Biodiesel as Renewable Fuel in Combustion Engines : A Review. Energies 2021;14:2565. doi:https://doi.org/10.3390/en14092565.

Response authors: This suggestion is outside the scope of this paper.

  1. The tables/figures inserted are not explained or discussed well in the text please discuss critically and explain all tables/figures in the text.

Response authors: We disagree with this statement and believe both tables and figures are discussed appropriately.

  1. Against which standard was the engine used homologated?

Response authors: This question seems not relevant to the paper. We wonder is there is a mix up of reviews?

  1. What are the tolerances for the emissions equipment? Cheaper multigas analyzers are notorious for having very high uncertainties and are generally not acceptable for academic publications.

Response authors: This question seems not relevant to the paper. We wonder is there is a mix up of reviews?

  1. There appears to be some sort of smoothing on the pressure data. How was this performed?

Response authors: This question seems not relevant to the paper. We wonder is there is a mix up of reviews?

  1. No error bars were put in the figures.

Response authors: This question seems not relevant to the paper. We wonder is there is a mix up of reviews?

  1. Conclusions of this manuscript lacks clear findings and future aspects. The authors are advised to write the conclusion comprehensively.

Response authors: This question seems not relevant to the paper. We wonder is there is a mix up of reviews?

  1. There are some grammatical issues, particularly in the introduction, and results & discussion section.

Response authors: This question seems not relevant to the paper. We wonder is there is a mix up of reviews?

  1. Validate the results section with the help of physicochemical properties not just with the previously published results.

Response authors: This question seems not relevant to the paper. We wonder is there is a mix up of reviews?

  1. Reduce the similarity to below 20%.

Response authors: This question seems not relevant to the paper. We wonder is there is a mix up of reviews?

  1. Between the lack of methodology and limited novelty, the authors have a significant amount of work to do to bring this paper up to a publishable standard.

Reviewer 3 Report

The authors have presented an interesting work with good methodology and results/analysis.

(i) However, the introduction is too short and needs to be strengthened. I will suggest that the authors considered reviewing more existing works on deterministic and probabilistic models to justify the need for their study. The novelty statement in the introduction also needs to be strengthened. What edge does this study have over other existing similar works? etc. 

Author Response

Response authors: We thank the reviewer for the feedback and have added to the introduction section along the lines suggested and added 15 new references to support this.

Round 2

Reviewer 1 Report

I think the authors have fixed the issues I raised last time.

I have only one question: Why you adopted triangle distribution for the values of e(vehicle, ICEV), but non-standard beta distribution for e(vehicle, BEV)? How do you choose the types of paramere distributions?

For the model itself, however, I guess it is fine for you to choose any type of distribution since you are using the Monte Carlo method.

Author Response

Response authors: The fitting procedure selects from a range of distributions discussed in the paper using the methods described in Section 2.3. In the case above the triangular distribution provided the best fit for e_vehICEV and the Beta distribution for e_vehBEV.

Reviewer 2 Report

Can be accepted.

Author Response

Response authors: Thank you.